# The SecM arrest peptide traps a pre-peptide bond formation state of the ribosome

Felix Gersteuer [1,4], Martino Morici [1,4], Sara Gabrielli [2], Keigo Fujiwara [3], Haaris A. Safdari[1], Helge Paternoga[1], Lars V. Bock[2], Shinobu Chiba [3] & Daniel N. Wilson [1] ✉

Nascent polypeptide chains can induce translational stalling to regulate gene expression. This is exemplified by the *E. coli* secretion monitor (SecM) arrest peptide that induces translational stalling to regulate expression of the downstream encoded SecA, an ATPase that co-operates with the SecYEG translocon to facilitate insertion of proteins into or through the cytoplasmic membrane. Here we present the structure of a ribosome stalled during translation of the full-length *E. coli* SecM arrest peptide at 2.0 Å resolution. The structure reveals that SecM arrests translation by stabilizing the Pro-tRNA in the A-site, but in a manner that prevents peptide bond formation with the SecM-peptidyl-tRNA in the P-site. By employing molecular dynamic simulations, we also provide insight into how a pulling force on the SecM nascent chain can relieve the SecM-mediated translation arrest. Collectively, the mechanisms determined here for SecM arrest and relief are also likely to be applicable for a variety of other arrest peptides that regulate components of the protein localization machinery identified across a wide range of bacteria lineages.

Cells have evolved elaborate post-transcriptional regulatory pathways to monitor and fine-tune expression of particular genes. One such strategy utilizes specific nascent polypeptide chains (NC) to induce translational arrest by inhibiting *in cis* the ribosome that is translating it. These so-called "arrest peptides" are usually encoded in upstream open reading frames (uORFs) where they induce translational stalling to regulate expression of a downstream gene[1–4]. Perhaps one of the best-characterized examples is the secretion monitor (SecM) arrest peptide that is involved in the regulation of the downstream *secA* gene in Gram-negative bacteria, such as *Escherichia coli*[2,5,6]. In the absence of SecM-mediated stalling, an intergenic stem-loop structure in the mRNA sequesters the ribosome-binding site (RBS) of the *secA* gene, preventing translation of the SecA protein (Fig. 1a). However, SecM-mediated stalling during translation of the *secM* uORF results in conformational changes within the mRNA that expose the downstream RBS and thereby promotes translation of the *secA* gene (Fig. 1b). SecA is an ATPase that functions together with the SecYEG protein-conducting channel to facilitate the targeting of secretory proteins into and through the cytoplasmic membrane[7–9]. Because *secM* encodes an N-terminal signal sequence (Fig. 1c), the SecM arrest peptide is itself a substrate for SecA. Importantly, the interaction of SecA with the N-terminal signal sequence of SecM as it emerges co-translationally from the ribosomal tunnel exerts a pulling force of the SecM NC that relieves the SecM-mediated translational arrest[2,5,10,11] (Fig. 1a). Thereby, an autoregulatory system is established such that when the intracellular levels of SecA are low, SecM stalling persists, resulting in the upregulation of the expression of *secA* (Fig. 1b). By contrast, as SecA levels are restored, SecM stalling is relieved, leading to repression in the expression of *secA*[2,6] (Fig. 1a).

[1]Institute for Biochemistry and Molecular Biology, University of Hamburg, Martin-Luther-King-Platz 6, 20146 Hamburg, Germany. [2]Theoretical and Computational Biophysics Department, Max Planck Institute for Multidisciplinary Sciences, Göttingen, Germany. [3]Faculty of Life Sciences and Institute for Protein Dynamics, Kyoto Sangyo University, Kamigamo, Motoyama, Kita-ku, Kyoto 603-8555, Japan. [4]These authors contributed equally: Felix Gersteuer, Martino Morici. ✉e-mail: Daniel.Wilson@chemie.uni-hamburg.de

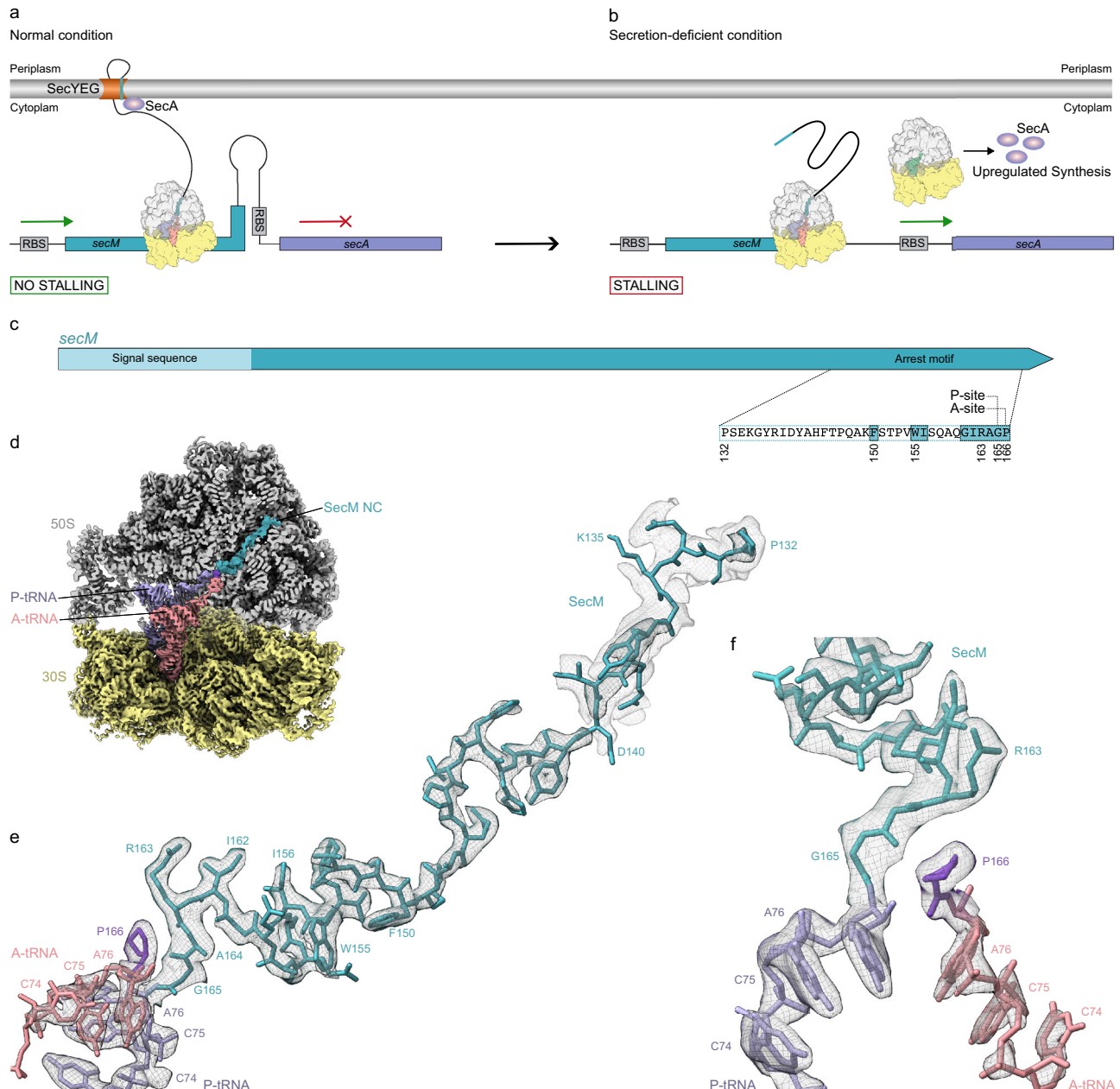

**Fig. 1 | Regulation of SecM and cryo-EM structure of SecM-SRC. a** Schematic representation of *secM-secA* mRNA illustrating the stem-loop structure at the stalling site of *secM* leader peptide (teal) that sequesters the ribosome-binding site (RBS) of the *secA* gene (lavender) thereby preventing *secA* translation. By SecYEG-mediated SecM translocation, SecM-induced stalling is relieved. **b** Upon stalling of the ribosome at the stalling site of *secM* (teal) the ribosome-binding site of the *secA* gene (lavender) becomes accessible and translation of *secA* starts. **c** Schematic representation of the SecM gene used in the SRC formation with SecM signal sequence and arrest motif as well as functionally relevant amino acids, A- and P-site of the arrest motif indicated. **d** Cryo-EM map of the 3D-refined *E. coli* SecM-SRC with transverse section of the 50S (grey) to reveal density for the nascent chain (teal), P-tRNA (lavender), proline 166 of SecM (grape), A-tRNA (salmon) and 30S (yellow). **e, f** Two views showing the cryo-EM map density (black mesh) for A- and P-site tRNA as well as the attached nascent chain and proline of the 3D refined *E. coli* SecM-SRC. The P-site tRNA (lavender) bears the SecM nascent chain (teal), whereas the A-site tRNA (salmon) carries proline (grape). Additional density at lower threshold for N-terminal part of nascent chain (grey mesh) in (**e**).

Biochemical studies have revealed that the SecM arrest peptide stalls the ribosome with the SecM NC attached to tRNA$^{Gly165}$ in the P-site and with Pro$^{166}$-tRNA in the A-site[12,13] (Fig. 1c). Alanine scanning mutagenesis identified residues Arg163 and Pro166 within the *E. coli* SecM sequence as being critical for SecM-mediated stalling, but also other residues that contribute to stalling, leading to designation of a SecM arrest motif $_{150}$FxxxxWIxxxxGIRAGP$_{166}$[14] (Fig. 1c). Although this small SecM arrest motif is sufficient to induce translational stalling, stronger arrest is observed when using the full-length SecM sequence, indicating that the regions N-terminal to the SecM arrest

motif also contribute to the stalling efficiency[14–16]. Biophysical studies proposed that SecM adopts a compacted conformation within the exit tunnel, and identified mutations (F150A, W155A and R163A) in SecM where compaction is maintained but stalling is reduced, suggesting that compaction is necessary but not sufficient to induce the translational arrest[17]. Mutations within 23S rRNA nucleotides, as well as alterations within ribosomal proteins, such as uL22, comprising the ribosomal tunnel reduce the efficiency of SecM-mediated stalling[14,18–21]. Specific SecM residues have also been crosslinked to uL22[22], collectively, suggesting that interaction between the SecM NC

and the ribosomal tunnel plays an important role in modulating the efficiency of stalling.

The first structures of SecM-stalled ribosomal complexes (SRC) were reported at 6–9 Å, leading to the proposal that SecM causes a shift in the position of the P-site tRNA, which interferes with peptide bond formation with the A-site Pro-tRNA[23]. A subsequent structural study[24] reported cryo-EM structures of SecM-SRC at higher resolution (3.3–3.7 Å), suggesting that SecM (i) induces conformational changes within the peptidyltransferase centre (PTC) that lead to an inactive state of the ribosome, and (ii) that the sidechain of the critically important Arg163 of the SecM sterically blocks the accommodation of Pro-tRNA in the A-site. The presence of a vacant A-site in the SecM-Gly-SRC structure[24] contrasted with previous biochemical studies indicating that the A-site is occupied by Pro-tRNA[12,13]. Also surprising was that the SecM NC was extended in the tunnel[24], rather than adopting a compacted conformation as suggested previously[17,25]. Of note, was that this SecM-SRC structure was determined using only 17 residues (150–166) of SecM, with the N-terminal residues being replaced by 2xStrep-TEV-tag, the N-terminal 40 residues of OmpA and a Myc-tag[24], therefore, the interaction of regions N-terminal of the arrest window could not be ascertained[15,16,26]. Finally, the SecM-SRC was purified in the presence of chloramphenicol, an antibiotic that inhibits elongation by binding to the PTC[27], and may therefore have also affected the final functional state that was visualized[24].

In addition to studies addressing the mechanism of SecM-mediated stalling, the SecM arrest peptide has been used extensively for generating ribosome-nascent chain complexes (RNCs) for functional studies[28–30], including ribosome display[31,32], real-time monitoring in vivo[33] and single molecule imaging[34,35], but particularly for investigating co-translational protein folding and targeting events[36–53]. Furthermore, molecular dynamics simulations based on available structural models for SecM have been performed to investigate how the pulling force could relieve the translational arrest[38,54–56]. Given the wide usage of SecM for diverse functional studies, it is important to understand the conformation of full-length SecM within the ribosomal tunnel, the number of the residues of SecM that transverse the ribosomal tunnel as well as the exact mechanism of action of SecM to inhibit translation elongation.

Here we report a cryo-electron microscopy (cryo-EM) structure of an *E. coli* SecM-stalled ribosomal complex (SRC) at 2.0 Å resolution. In contrast to the previous SecM-RNC[24], we observe one major functional state of the ribosome bearing a SecM-peptidyl-Gly-tRNA in the P-site and Pro-tRNA in the A-site. Our structure reveals that SecM stalls translation by interfering with peptide bond formation, rather than preventing accommodation of the A-site tRNA, as suggested previously[24]. Specifically, our data support a model whereby interactions between the $_{163}$RAG$_{165}$ motif in the SecM nascent chain attached to the P-site tRNA with the Pro$^{166}$-tRNA in the A-site prevent the proton transfer necessary to allow the nucleophilic attack during peptide bond formation. Additionally, we observe the formation of a short seven amino acid (aa) α-helix encompassing residues Thr152 and Glu158 of SecM. As a consequence, the change in register leads to a completely different set of interactions between the SecM NC and tunnel components compared to previous reports[23,24]. We believe that the knowledge that 40, rather than 30, residues of SecM are accommodated within the exit tunnel will also be of general importance when using SecM-SRC to investigate co-translational folding and targeting events.

## Results

### Cryo-EM structure of *E. coli* SecM-SRC at 2.0 Å resolution

*E. coli* SecM-stalled ribosome complexes (SecM-SRC) were generated using a fully reconstituted *E. coli* in vitro translation system and purified with the N-terminal FLAG affinity tag (see "Methods"). Unlike previous SecM-SRC structures that employed a SecM arrest window of

**Table 1 | Cryo-EM data collection, refinement and validation statistics**

| Model | SecM-SRC (P-tRNA, A-tRNA) |
|---|---|
| EMDB ID | 18534 |
| PDB ID | 8QOA |
| Data collection and processing | |
| Magnification (×) | 105,000 |
| Acceleration voltage (kV) | 300 |
| Electron fluence (e⁻/Å²) | 40 |
| Defocus range (μm) | −0.3 to −0.9 |
| Pixel size (Å) | 0.83 |
| Initial particles | 398,692 |
| Final particles | 300,107 |
| Average resolution (Å) (FSC threshold 0.143) | 2.0 |
| Model composition | |
| Initial model used (PDB code) | 7K00 |
| Atoms | 147,754 |
| Protein residues | 5607 |
| RNA bases | 4549 |
| Refinement | |
| Map CC around atoms | 0.72 |
| Map CC whole unit cell | 0.71 |
| Map sharpening B factor (Å²) | −39.79 |
| R.M.S. deviations | |
| Bond lengths (Å) | 0.008 |
| Bond angles (°) | 1.529 |
| Validation | |
| MolProbity score | 0.86 |
| Clash score | 0.37 |
| Poor rotamers (%) | 0.65 |
| Ramachandran statistics | |
| Favoured (%) | 96.50 |
| Allowed (%) | 3.39 |
| Outlier (%) | 0.11 |
| Ramachandran Z-score | −1.37 |

17–27 residues[23,24], we utilized the full-length wildtype *E. coli* SecM sequence comprising the full 170 residues (Fig. 1c). Moreover, in contrast to previous structural studies on SecM-SRC[23,24], the antibiotic chloramphenicol was not added during any stage of the sample preparation. The purified SecM-SRC was applied to cryo-EM grids and analyzed using single particle cryo-EM. A total of 4388 micrographs were collected on a Titan Krios G3i equipped with a K3 direct electron detector, yielding 398,692 particles after 2D classification (Supplementary Fig. 1 and Table 1). Focused 3D classification on the 377,762 particles containing 70S ribosomes revealed one major class of non-rotated 70S ribosomes bearing A- and P-site tRNAs (75%; 300,107 particles) as well as one minor class with rotated 70S ribosomes with hybrid A/P- and P/E-site tRNAs (9,1%; 36,489 particles), collectively representing a total of 84% of the initial ribosomal particles (Supplementary Fig. 1). The 70S ribosome with A- and P-site tRNAs was further refined, yielding a cryo-EM map of the SecM-SRC with an average resolution of 2.0 Å (Fig. 1d, Table 1 and Supplementary Fig. 1 and 2). In the SecM-SRC, density for the SecM nascent polypeptide chain (NC) was observed throughout the ribosomal exit tunnel (Fig. 1d), enabling 34 amino acids (residues Pro132 to Gly165) of SecM to be modelled (Fig. 1e). With the exception of the four residues (Pro132-Lys135) near the tunnel exit, the density was well-resolved enabling unambiguous placement of almost all the sidechains (Fig. 1e, Supplementary Fig. 2,

Supplementary Movie 1), especially the C-terminally conserved $_{163}$RAG$_{165}$ motif that is directly linked to the CCA-end of the P-site tRNA (Fig. 1f). In addition, the high quality of the cryo-EM density map allowed the Pro$_{166}$ moiety attached to the CCA-end of the A-site tRNA to be unambiguously identified and modelled (Fig. 1f, Supplementary Movie 1). We also subsorted and refined the rotated SecM-SRC population with hybrid A/P- and P/E-site tRNAs, yielding a cryo-EM map generated from 36,489 particles with average resolution of 2.6 Å (Supplementary Figs. 1 and 3). Although the ribosome was well-resolved, the density for the tRNAs and the NC were less defined, precluding a molecular model to be generated (Supplementary Fig. 3). Overall the SecM NC path seems similar to that observed in the non-rotated SecM-SRC (Supplementary Fig. 3), therefore, we presume this state represents a small population of SecM-SRC that has undergone peptide bond formation during the long purification process (>4 h at 4 °C, see "Methods"), such that the deacylated tRNA$^{Gly}$ is present in the P/E site and the peptidyl-SecM-Gly-Pro-tRNA$^{Pro166}$ is now shifted into the A/P-site, as observed previously[23,24]. However, we cannot exclude that this population represents a mixture of states, which coupled with the poor resolution of NC, meant that state was not analyzed further. Taken together, the cryo-EM structure of the SecM-SRC revealed that the majority of ribosomes bear the SecM-Gly-tRNA in the P-site and have Pro-tRNA in the A-site, consistent with previous biochemical analysis[12,13]. This supports the suggestion that the SecM NC interferes with peptide bond formation between the peptidyl-RAG-tRNA in the P-site and the incoming A-site Pro-tRNA[12,13].

## The SecM NC adopts a helical structure within the NPET

The path of the SecM NC is observed from the PTC, where the C-terminus is attached to the tRNA$^{Gly}$, throughout the tunnel to the vestibule where the tunnel widens at the exit (Fig. 2a, b). The SecM NC makes no stable contact with uL4 as it passes through the constriction, whereas multiple interactions with uL22 are observed, not only at the constriction, but also deeper in the tunnel (Fig. 2b). The N-terminal residues Pro132-Lys135 of SecM are within close proximity of uL23 (Fig. 2b), but do not appear to directly make contact. While the majority of the SecM NC adopts an extended conformation, the region located directly between the PTC and the constriction is clearly compacted (Fig. 2a, b). This is consistent with secondary structure predictions of the SecM NC that suggest a high probability of α-helical formation within this region (Fig. 2c). Careful inspection of the molecular model of the SecM NC within this region indeed revealed that for residues Glu158 to Thr152 of SecM adopt a standard [i + 4 → i] α-helix where each backbone nitrogen (N-H) forms a hydrogen bond with the backbone carbonyl-oxygen (C=O) of the amino acid four residues earlier (Fig. 2d, e). The seven-residue α-helix forms despite the presence of Pro153, which does not break the helix but its location at the N-terminus may rather facilitate its formation[57] (Fig. 2d, e). Although reminiscent of the ten residue α-helix that was observed at the PTC of the VemP arrest peptide[58], the location of the SecM α-helix is shifted by 12 Å deeper into the tunnel (Supplementary Fig. 4a–c), such that, unlike VemP[58], the SecM α-helix does not perturb the conformation of 23S rRNA nucleotides at the PTC (see later). Rather the location is more similar to the helical regions observed in the exit tunnel of the TnaC and hCMV arrest peptides structures[59,60] (Supplementary Fig. 4d–g).

The presence of an α-helical conformation for residues Thr152-Glu158 of SecM observed in the SecM-SRC structure determined here (Fig. 2f) is in excellent agreement with compaction observed in a previous study measuring florescence resonance energy transfer (FRET) between the acceptor and donor probes at positions 135 and 159 of SecM, respectively[17]. Compared to a theoretical fully extended conformation spanning 3.5 Å per residue, the FRET study predicted a compaction of 2.6 Å per residue, which compares well with the 2.0 Å per residue observed here (distance between positions 135 and 159 of 50 Å/25 residues). By contrast, no compaction was observed in the

previous SecM-SRC structure[24] and therefore the last NC residue modelled was Glu139 that forms part of the c-Myc tag and was equivalent to Ile139 of SecM (Fig. 2g). Because of the difference in the degree of compaction between the SecM-SRC determined here (Fig. 2f) and the previous SecM-SRC structure (PDB ID 3JBU)[24] (Fig. 2g), the register of the SecM residues spanning the ribosomal exit tunnel is completely different (Fig. 2h). This is exemplified by Phe150 of SecM, which in the previous structure[24] was located deep in the tunnel past the uL4-uL22 constriction site, whereas in the SecM-SRC structure determined here, Phe150 is located 25 Å away on the PTC side of the constriction site (Fig. 2h). The compacted conformation observed in the SecM-SRC structure determined here is also more consistent with biochemical data reporting crosslinking between Tyr141 of SecM and uL22, as well as the lack of crosslinking between residues 149 and 152 of SecM and uL22 (Fig. 2f)[22].

## Interaction of SecM with ribosomal proteins of the NPET

As mentioned, a consequence of the compacted conformation of SecM is that the residues (Phe150, Trp155, Ile156, $_{161}$GIRAGP$_{166}$) encompassing the SecM motif (FxxxxWIxxxxGIRAGP) are all located before the constriction (Fig. 2f), rather than forming direct interactions with ribosomal proteins uL4 and uL22 located deeper in the tunnel, as proposed previously[24]. Instead, in the SecM-SRC determined here, we observe that the residues N-terminal to Phe150, specifically, residues 132–149, of SecM form multiple interactions with both ribosomal protein and rRNA components of the exit tunnel. This is in excellent agreement with previous studies indicating that regions N-terminal to the SecM arrest window within the full-length SecM also contribute to the efficiency of stalling[14–16,26]. Although the N-terminal region of SecM at the exit tunnel site is poorly resolved, the density for the NC clearly passes between the tip of uL23 and 23S rRNA helix 50 (H50), with strong density suggesting that Pro132 of SecM forms stacking interactions with the nucleobase of A1321 within H50 (Fig. 3a). The density for the residues Tyr137 to Lys149 of SecM is well-resolved (Figs. 1e and 2b), presumably due to the multiple interactions observed with tunnel components, in particular, uL22 (Fig. 3a–c). Briefly, the sidechain of Tyr137 of SecM comes within hydrogen bonding distance of the backbone of Ile85 of uL22 and can form stacking interactions with Arg84 of uL22 (Fig. 3a). Direct hydrogen bonds are also possible from the sidechain of Tyr141 of SecM and the backbone of Gly91 of uL22, as well as additional interactions between the backbone of His143 and Gln147 of SecM with the backbone of Lys90 and Ala93 of uL22 (Fig. 3b). The high quality of the map enables a network of water-mediated interactions to be described involving residues (Ala142, Thr145, Pro153) of the SecM NC with residues (Lys90, Arg92 and Ala93) of uL22 as well as 23S rRNA nucleotides A1614 and A751 (Fig. 3c). Interaction with A751 likely explains why the insertion of an adenine within the 5-adenine stretch between A749-A753 reduces SecM-mediated stalling, albeit resulting in a relatively minor effect[14,19].

There is good agreement between the interactions observed between SecM and uL22 in the SecM-SRC and previously reported alterations in uL22 that reduce the stalling efficiency of SecM[14,18]. This includes, for example, substitutions at residues Gly91, Ala93 or Arg84, insertions (+2 and +15 aa at position 99 and 105, respectively) within the loop of uL22, as well as deletion of the $_{82}$MKR$_{84}$ motif or the entire loop in uL22[14,18]. In most cases, the alterations would be predicted to perturb the interactions between SecM and uL22 by either directly introducing steric clashes (substitutions and insertions) and/or inducing conformational changes in the uL22 loop (insertions and deletions) (Supplementary Fig. 5a–f). The later scenario is exemplified by the structures of ribosomal 50S subunits with insertions or deletions in uL22 that lead to dramatic rearrangements in the loop of uL22[61,62], which would be incompatible with the observed path of the SecM NC (Supplementary Fig. 5g–j). By contrast, we observe no defined interaction between the SecM NC and uL4, with the closest point of contact

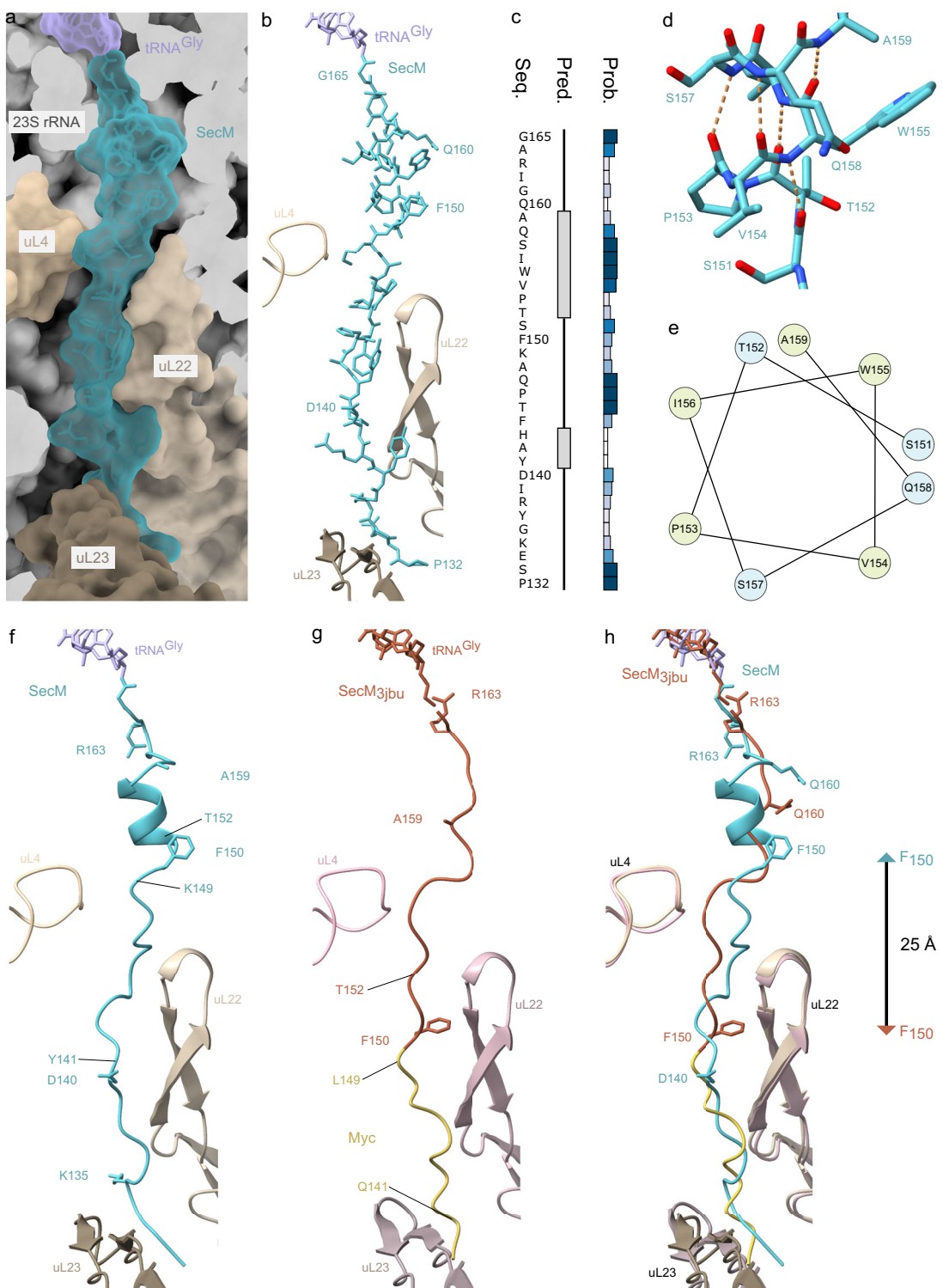

**Fig. 2 | Formation of an α-helix inside the NPET by the SecM peptide.**
**a** Transverse section of the NPET shown as surface (grey) with P-tRNA (lavender) and SecM (teal) in relation to uL4 (light gold), uL22 (gold) and uL23 (dark gold) in surface representation. **b** Cryo-EM density (transparent teal, threshold 0.008/~2.6 σ) and model of SecM (teal) attached to the P-tRNA (lavender) in relation to uL4 (light gold), uL22 (gold) and uL23 (dark gold). **c** Secondary structure prediction (Pred.) and probability (Prob.) of the SecM (Seq.) inside the NPET determined using PSIPRED. **d** Helix region of SecM (teal) inside the NPET and potential hydrogen bonds shown as dashed orange lines. **e** Downward cross-sectional view of the SecM helix axis with non-polar amino acids coloured in yellow and polar amino acids coloured in blue. **f** Structure of SecM (teal) in ribbon representation attached to the P-tRNA (lavender) in relation to uL4 (light gold), uL22 (gold) and uL23 (dark gold). **g** Myc-SecM (PDB ID 3JBU)[24] attached to the P-tRNA (tangerine/yellow) in relation to uL4 (light rose), uL22 (rose) and uL23 (dark rose). **h** Overlay (aligned on basis of 23S rRNA) of (**f**) SecM and (**g**) SecM$_{3jbu}$ with distance between F150 position from the two models arrowed.

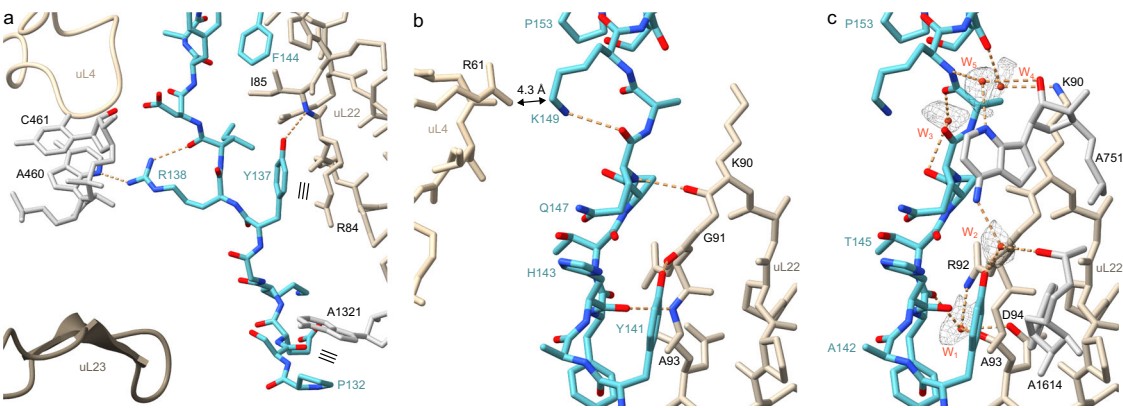

**Fig. 3 | Interactions of SecM with components of the NPET.** Interactions of **a** N-terminal and **b**, **c** middle part of SecM (teal) inside the NPET with 23S rRNA (grey), uL4 (light gold), uL22 (gold) and uL23 (dark gold). In (**a**) and (**b**) direct interactions are shown whereas in (**c**) water-mediated interactions for the middle part of SecM (teal) are indicated. Potential hydrogen bonds are shown as dashed orange lines, stacking interactions as three parallel lines and water molecules as red spheres with meshed density.

being 4.3 Å between the sidechains of Lys149 of SecM and Arg61 of uL4 (Fig. 3b). Consistently, selection of mutants that relieve SecM-mediated stalling were identified only in uL22, but not in uL4[14], and engineered uL4 substitutions at positions 62, 63 and 66 had no effect on SecM arrest[18]. The exception is an insertion of six amino acids at position 72 of uL4 that was reported to have a minor effect on SecM pausing[18], however, Ser72 of uL4 is located ~20 Å away from the SecM NC and therefore any effects are likely to be indirect via conformational changes in uL4, analogous to the uL22 loop insertions and deletions. We note that the sidechain of Arg138 of SecM can form a hydrogen bond with 23S rRNA nucleotide A460 located in H23 (Fig. 3a), and that H23 also contacts uL4, raising the possibility that this insertion in uL4 indirectly effects SecM stalling via perturbing H23.

A recent study generating deletions in the N-terminus of SecM revealed that residues 58–98 of SecM contribute to the efficiency of SecM-mediated stalling[16]. Because this region of SecM was predicted to adopt an α-helical secondary structure and Tyr80 located within this α-helix was shown to crosslink to uL23, the authors proposed that the interaction is likely to occur outside the tunnel exit[16]. Since we used the full-length SecM sequence for the SecM-SRC, we carefully analyzed the tunnel exit site and indeed discovered an additional density located in proximity to uL23, albeit only visible at low threshold levels (Supplementary Fig. 6). Although the additional density would be consistent with an α-helical structure, the density is poorly resolved, precluding a molecular model to be generated. Moreover, the lack of density connecting the SecM NC within the tunnel with the helical density at the tunnel exit makes it difficult to assign this region to any specific part of the N-terminus of SecM. In fact, we cannot rule out that the additional density actually represents the N-terminal signal sequence of SecM, which was also included in our construct, although we note that the binding position differs from that reported previously for signal sequences bound to ribosomal complexes[63,64] (Supplementary Fig. 6).

### Interaction of SecM with 23S rRNA nucleotides of the NPET

As mentioned above, a surprise from the SecM-SRC structure determined here was that in contrast to the previous SecM-SRC structure[24], the residues of the SecM motif (FxxxxWIxxxxGIRAGP) are not spread out through the exit tunnel, but are rather located in the upper third of the tunnel, at or adjacent to the PTC (Fig. 4a). The α-helical conformation of this region of SecM coupled with the hydrophobic nature of many of the residues suggests that many of the contacts with the ribosome utilize van der Waals interactions (Fig. 4b). This is exemplified by the interaction with A2058, which is surrounded by the sidechains of Thr152, Pro153, Ile156 and Ile162 of SecM (Fig. 4a and

Supplementary Fig. 7a). This intimate interaction explains why the A2058G mutation, which would lead to a clash with the sidechains of Thr152 and Ile156 of SecM (Supplementary Fig. 7b), relieves SecM-mediated stalling[14,19]. Conversely, dimethylation of A2058 does not affect SecM-mediated stalling[18] and no clash would be predicted based on the structure of the SecM-SRC (Supplementary Fig. 7c). We note that eukaryotic ribosomes contain G3904 in the position equivalent to *E. coli* A2058 (Supplementary Fig. 7d), consistent with our findings that SecM stalling does not work on eukaryotic ribosomes (see below).

Additional direct and water-mediated hydrogen bonds as well as stacking interactions are also observed that are likely to contribute to SecM stalling by stabilizing a defined conformation of the SecM NC. Specifically, the sidechain of Phe150 is observed to stack on the nucleobase of U2609 (Fig. 4a), and mutation of either Phe150[14] or U2609[19] has been reported to reduce the efficiency SecM-stalling. Direct hydrogen bonds are possible between the sidechain of Ser157 of SecM and the ribose of A2062, as well as the backbone carbonyls of Gly161 and Ile162 with the nucleobases of U2506 and A2062, respectively (Fig. 4c). Consistently, mutation of A2062U, which would lead to a loss of interaction with Ile162 (Supplementary Fig. 7e, f), reduces SecM-mediated stalling[20]. The same study also demonstrated that A2503G mutations reduce SecM-mediated stalling[20], although we observe no direct interaction between A2503 and the SecM NC (Supplementary Fig. 7g). Instead, the A2503G mutation may induce an alternative conformation of A2062[20] that is incompatible with the modelled path of SecM (Supplementary Fig. 7h).

Additionally, potential water-mediated interactions link Ser157 of SecM with A2062, and Gln160 and Ala159 with U2585 (Fig. 4d), which are likely to contribute to stabilizing the observed conformation of the SecM NC. However, the most intricate network of interactions is observed for Arg163 of SecM, which inserts into a pocket formed by 23S rRNA nucleotides G2061 and A2503-U2506 (Fig. 4e, f). Within the binding pocket, Arg163 stacks upon Ψ2504 (Fig. 4a) and can potentially establish seven hydrogen bonds with 23S rRNA nucleotides, five direct interactions as well as two mediated via water molecules (Fig. 4f). The interactions are likely to be critical for SecM mediated stalling since Arg163 was reported to be one of only three amino acids positions (together with Ile162 and Pro166) that was invariant in all sequenced SecM homologues[22], and mutation of Arg163 (as well as Pro166 in the A-site) produced the strongest relief of SecM-mediated stalling[14].

In eukaryotic ribosomes, U4450, the equivalent nucleotide to EcΨ2504, adopts a different conformation that would prevent the stacking interaction with Arg163 (Supplementary Fig. 8a), which may

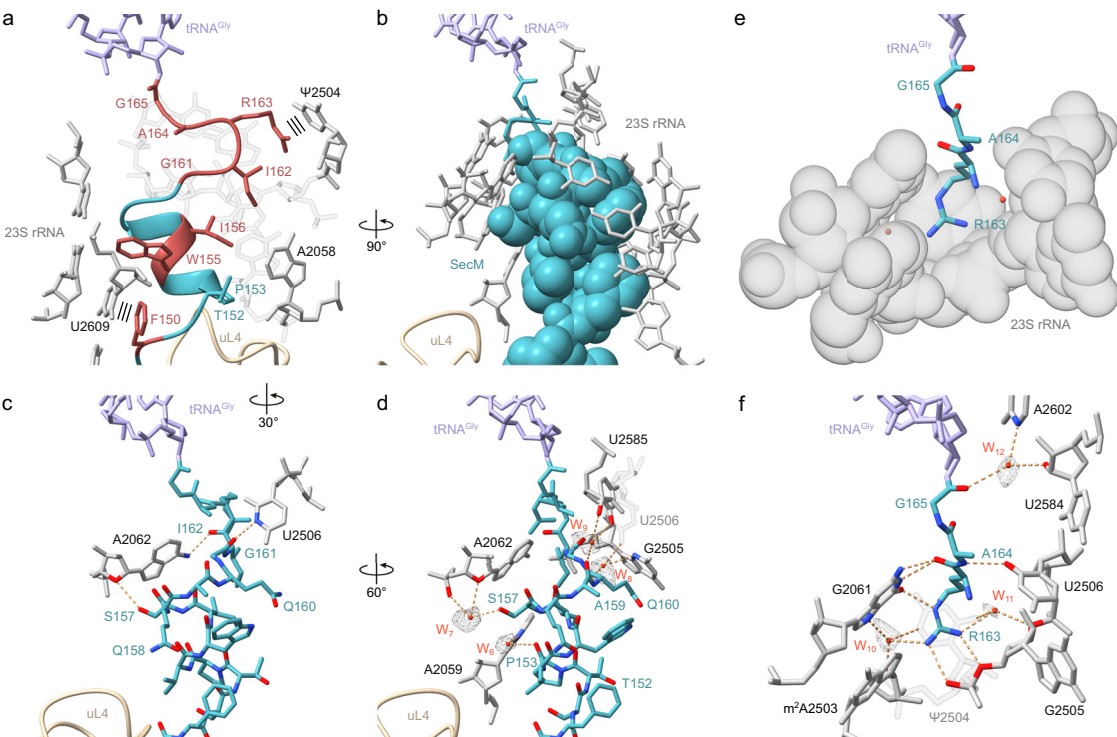

**Fig. 4 | Interactions of SecM stalling motif with components of the NPET.**
**a** SecM arrest peptide (teal) with surrounding 23S rRNA (grey) and residues of the SecM stalling motif highlighted in red. Stacking interactions depicted as three parallel lines. **b** Space filling representation of SecM stalling motif (teal) from $F_{150}$ to $I_{162}$ with surrounding 23S rRNA (grey). **c** Direct interactions of SecM stalling motif (teal) from $F_{150}$ to $I_{162}$ with 23S rRNA (grey). **d** Water-mediated interactions of SecM stalling motif (teal) from $F_{150}$ to $I_{162}$ with 23S rRNA (grey). **e** Space filling representation of a distinct pocket of the NPET formed by 23S rRNA (grey) and entering SecM $_{163}$RAG$_{165}$ (teal). **f** Direct and water-mediated interactions of SecM $_{163}$RAG$_{165}$ (teal) with 23S rRNA (grey). Potential hydrogen bonds are shown as dashed orange lines and water molecules as red spheres (with meshed density).

prevent stalling in eukaryotes, although to our knowledge this is not known. To test this, we introduced the residues of SecM, as well as the Pro166Ala variant, into a GFP-LacZ reporter and monitored for the presence of peptidyl-tRNA and full-length protein after incubation in a rabbit reticulocyte in vitro translation system (Supplementary Fig. 8b). As a positive control, we employed an arrest-enhanced variant (S255A) of the XBP1u arrest peptide, where stalling was observed as the accumulation of peptidyl-tRNA that is resolved upon RNase treatment (Supplementary Fig. 8b), as expected[65]. By contrast, we observed no accumulation of SecM-peptidyl-tRNA (Supplementary Fig. 8b), suggesting that SecM does not mediate efficient translation arrest on eukaryotic ribosomes.

## SecM stabilizes a pre-attack state of the PTC

In the previous structure of SecM-SRC, Arg163 was modelled with the sidechain extending into the A-site where it would sterically block accommodation of the Pro-tRNA[24]. However, in the structure presented here, the sidechain of Arg163 of SecM is oriented differently (Fig. 2f, g) such that it would not interfere with Pro-tRNA accommodation at the A-site of the PTC. To understand how SecM allows accommodation of Pro-tRNA at the A-site, but prevents peptide bond formation with the SecM-peptidyl-tRNA, we compared the PTC of the SecM-SRC with that of pre-attack state ribosomal complexes[66,67] (Fig. 5a–c). In the pre-attack state, the α-amino group of the A-site Phe-tRNA is positioned ~3.0 Å from the carbonyl-carbon of the peptidyl-tRNA in the P-site, but peptide bond formation cannot occur because the peptide is linked to the P-site tRNA with an amide, rather than an ester, linkage[66,67] (Fig. 5a). In the SecM-SRC, the Pro-tRNA is accommodated in the A-site and the nitrogen of the Pro166 moiety on the A-site tRNA is located ~4.3 Å from the carbonyl-carbon of the SecM-peptidyl-tRNA in the P-site, yet peptide bond formation has not

occurred, even though the SecM peptide is linked to the P-site tRNA by an ester linkage (Fig. 5b). The Pro-tRNA appears to be fully accommodated in the A-site since the conformation of the 23S rRNA nucleotides at the PTC is indistinguishable from that observed in the pre-attack state structures[66,67] (Supplementary Fig. 8c, d), indicating that the induced conformation that is concomitant with A-site tRNA accommodation has been attained. Moreover, superimposition of the pre-attack state[66,67] and the SecM-SRC reveals an identical placement (within the limits of the resolution) of the CCA-end of the A-site tRNA (Fig. 5c). The increased distance between the A-site nitrogen and P-site carbonyl-carbon in the SecM-SRC appears to result from both a shifted path (by ~0.9 Å) of the SecM NC in the P-site as well as a different position (by ~0.7 Å) of the nitrogen (secondary amine) in the proline moiety, as compared to the nitrogen (primary amine) in other amino acids, such as Phe[66,67] (Fig. 5c). Although the distance is larger in SecM compared to the pre-attack state, it remains unclear whether this would be sufficient to effectively prevent the nucleophilic attack required for peptide bond formation to occur.

For peptide bond formation to occur, a proton needs to be extracted from the α-amino group of the amino acid linked to the A-site tRNA. In current models for peptide bond formation[66,68–70], this is performed by the 2′ OH of A76 of the P-site tRNA, which subsequently increases the nucleophilicity of the α-amino group, thereby facilitating the nucleophilic attack of the lone pair electrons onto the carbonyl-carbon of the first amino acid linked to the P-site tRNA (Fig. 5d). In principle, the same pathway should be employed for amino acids such as proline with a secondary amine, with the major difference being the presence of only a single hydrogen on the nitrogen of proline (Fig. 5e), rather than two hydrogens for amino acids with primary amines (Fig. 5d). It is important to emphasize that the resolution of the SecM-SRC (and to date any other ribosomal complexes) is currently

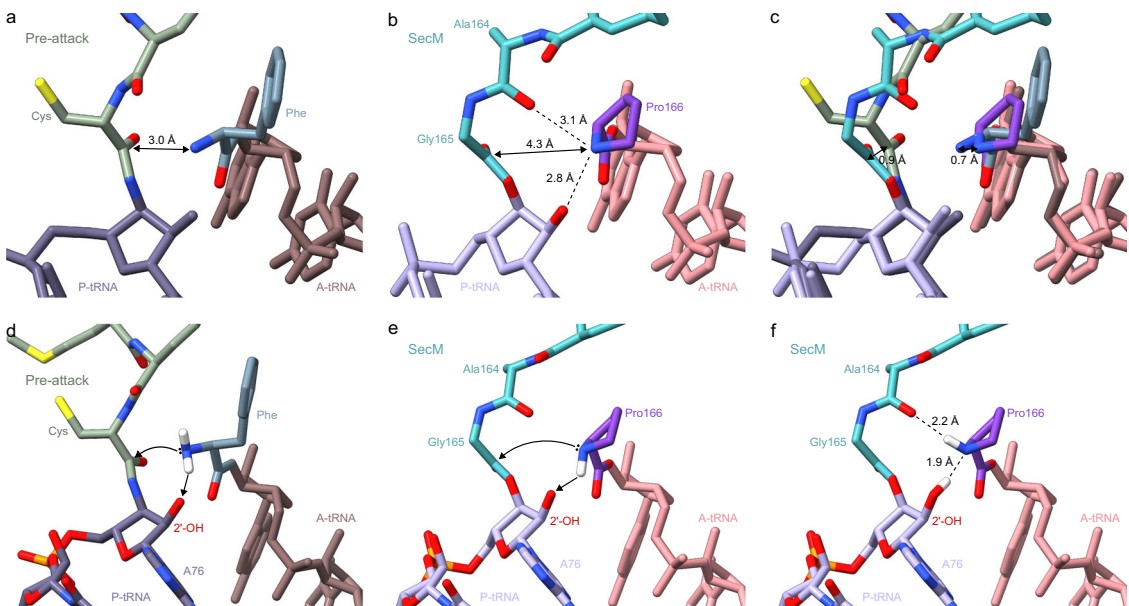

**Fig. 5 | Model for SecM-mediated PTC arrangement leading to translational stalling. a** View of the PTC of a pre-attack state (PDB ID 8CVK)[67], showing a tripeptidyl-NH-tRNA (green/dark lavender) at the P-site and a phenyl-NH-tRNA (slate blue/brown) at the A-site. The distance between the attacking amine of the A-tRNA and the carbonyl carbon of the P-site is indicated by a double arrow. **b** Same view as (**a**), but for SecM-SRC with SecM-tRNA (teal/lavender) in the P-site and Pro-tRNA (grape/salmon) in the A-site. **c** Overlay of (**a**) and (**b**) (aligned on the basis of 23S rRNA) highlighting the difference in the distance between the attacking amino groups at the A-site and the carbonyl carbon at the P-site. **d** Schematic view of the PTC of a pre-attack from (**a**), but with hydrogen atoms (white) modelled in silico for the amino group of the phenylalanine in the A-site. Black spheres indicate the lone pair electrons that make the nucleophilic attack (arrowed) on the carbonyl carbon of the cysteine attached to the P-site tRNA. **e** Same schematic as (**b**), but with hydrogen atom (white) modelled in silico towards the 2'OH of A76 of the P-site tRNA, which would allow a nucleophilic attack (arrowed) on the carbonyl carbon of the Gly165 attached to the P-site tRNA. **f** Same schematic as (**e**) but with the hydrogen atom (white) modelled towards the carbonyl of Ala164, a conformation that would prohibit any nucleophilic attack.

insufficient to observe hydrogens directly, and therefore their position can only be predicted or inferred via hydrogen bonding interactions. Indeed, careful examination of the environment of the nitrogen of the A-site Pro166 in the SecM-SRC reveals that the nitrogen is not only in hydrogen bonding distance to the 2' OH of A76 (2.8 Å in Fig. 5b), as expected, but also to the carbonyl-oxygen of Ala164 (3.1 Å in Fig. 5b), suggesting that these two hydrogen bonds are present simultaneously. Because the carbonyl oxygen of Ala164 can only act as an hydrogen bond acceptor, this suggests that the sole hydrogen of Pro166 must be donated to allow this bond to form (Fig. 5f). A consequence of this is that the 2' OH of A76 must also act as a donor to enable the second hydrogen bond, which would then form with the lone pair electrons on Pro166 (Fig. 5f). Thus, the hydrogen bonding pattern for Pro166 in the SecM-SRC completely disfavours peptide bond formation because (i) the 2' OH of A76 acts as a donor, rather than extracting a proton from Pro166 to increase the nucleophilicity, and (ii) the hydrogen bond with the carbonyl-oxygen of Ala164 creates a geometry where the lone pair electrons cannot make a nucleophilic attack on the carbonyl-carbon of the Gly165 on the P-site tRNA (Fig. 5f). We note that in the pre-attack state, the carbonyl-oxygen of the equivalent amino acid to Ala164 is oriented differently and further away from the A-site nitrogen, but that even if a hydrogen bond could form, the presence of two hydrogens on the primary amine of such an amino acid in the A-site would still allow extraction of a proton by the 2'OH of A76 while allowing an optimal geometry for peptide bond formation to be attained (Fig. 5d). This is also likely to explain why Pro166, bearing a secondary amine, is critical for SecM-mediated stalling and mutations to any amino acid having a primary amine, such as alanine[14,22], but also serine, histidine or arginine[71], lead to relief of stalling.

## Relief of stalling by pulling on the N-terminus of SecM

SecM stalling is released in vivo by a mechanical pulling force caused by interaction of the N-terminal signal sequence of SecM with SecA[2,5,10,11]. To investigate how this pulling force relieves translational stalling and how this is influenced by the presence of the α-helix in the tunnel, we performed all-atom explicit-solvent molecular dynamics (MD) simulations of the SecM-SRC. Two sets of simulations were carried out: unbiased simulations in the absence of a pulling force and pulling simulations where a harmonic spring potential acts on the N-terminal Pro132 of the SecM NC. During the pulling simulations, the spring position was moved by 56 Å in the direction of the tunnel axis with a constant velocity, exerting a force on Pro132. To check if the observed order of events depends on the pulling velocity, we carried out sets of 8 independent simulations with pulling times τ ranging between 32 ns and 1024 ns. Throughout the unbiased simulations, we observed the SecM α-helix to remain very stable and the fluctuations (rmsf) of the SecM residues to be small (Fig. 6a). In agreement with the cryo-EM structure where N-terminal residues were less well-resolved (Fig. 1e), we observe increased fluctuations for these residues (Fig. 6a). When pulling on the N-terminus, the α-helix could either remain folded and be pulled through the constriction as a whole, or it could unfold before passing through the constriction. The positions of SecM residues along the tunnel during the slower simulations showed that, in the beginning (0–384 ns), the N-terminal part straightens while the C-terminal part remains in place (Fig. 6b and Supplementary Movie 2). When the extension reaches the helix, it unfolds in a step-wise manner starting from the N-terminal side (384–640 ns). The unfolding of the α-helix before reaching the constriction site suggests that the constriction site acts as barrier for α-helices. Only after the helix is completely unfolded (768–1024 ns), can Ala164 of SecM shift away from the positions observed in the unbiased simulations, such that nitrogen of Pro166 cannot form a hydrogen bond with the carbonyl-oxygen of Ala164 anymore (Fig. 6c), thereby providing a rationale for the relief of stalling. This order of events was observed in all simulations and the N-terminus positions at which they occur were very similar (Fig. 6d). These observations were independent of the pulling velocities used in

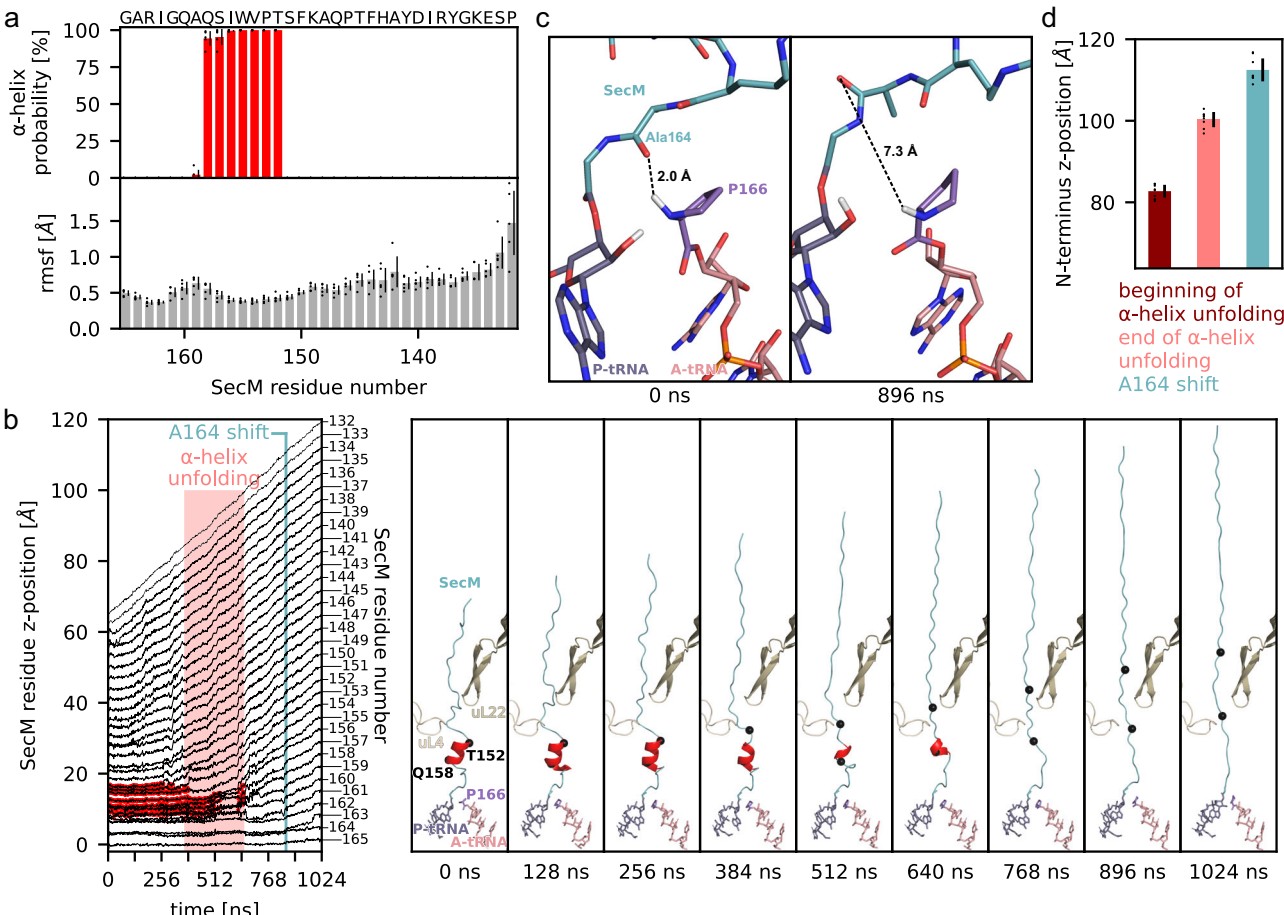

**Fig. 6 | MD simulations of the stalling release by pulling on N-terminus.**
**a** Probability of SecM residues being in an α-helix and their root mean square fluctuations in the absence of a pulling force. Mean (bars) and standard deviations (black lines) are shown for 5 independent simulations (circles). **b** Left panel: For one pulling simulation (length 1024 ns), positions of SecM residues along the tunnel axis are shown as a function of time. Initial G165 position is set to zero. Residues in α-helix secondary structure are highlighted in red. Unfolding of α-helix and beginning of A164 shift are indicated by light red rectangle and teal vertical line,

respectively. Right panel: intermediate structures at indicated times.
**c** Conformation of the PTC before pulling and after the A164 shift. Distance between Pro166 and A164 carbonyl oxygen. **d** Mean and standard deviation of N-terminus position at beginning (dark red) and end of α-helix unfolding (light red) as well as A164 shift (teal) are shown for 8 independent simulations (circles). DSSP[104] was used to assign α-helices. Source data be obtained from Zenodo (10.5281/zenodo.10492465).

the simulations (Supplementary Fig. 9a), suggesting that the helix also unfolds first in vivo. In the simulations, the maximum force during unfolding was always higher than before unfolding, indicating that helix unfolding represents the first barrier encountered during pulling (Supplementary Fig. 9b). This observation is consistent with the stability of the helix determining the force required to release the stalling, rendering it crucial for the fine-tuning of the stalling relief mechanism.

## Discussion

The cryo-EM structure of an *E. coli* ribosome stalled during translation of the full-length *E. coli* SecM sequence at 2.0 Å resolution allows the mechanism by which SecM induces translational arrest to be completely revised (Fig. 7). In our model, the SecM arrest peptide stalls the elongating ribosome in a pre-peptide bond formation state with SecM-peptidyl-Gly-tRNA in the P-site and Pro166-tRNA in the A-site (Fig. 7a). The structure suggests that the accommodation of the Pro-tRNA at the A-site is not affected, but rather that the SecM-peptidyl-tRNA actually stabilizes the Pro-tRNA in the A-site by interacting directly with the Pro moiety (Fig. 7a). Specifically, we observe that the carbonyl-oxygen of Ala164 comes within hydrogen bonding distance and geometry to the nitrogen of the A-site proline. Because Pro is the only natural amino acid with a secondary amine, the hydrogen bond formed between Pro166 and Ala164 sequesters the single hydrogen of Pro166 and

thereby prevents extraction of this proton by the 2′ OH of A76 of the P-site tRNA. Instead, we suggest that the 2′ OH actually donates a proton to form a hydrogen bond with the lone-pair electrons of the nitrogen on Pro166. Collectively, this creates a chemical environment and geometry that disfavours the nucleophilic attack necessary for a peptide bond formation to occur (Fig. 7a). Importantly, this model rationalizes why Pro166 is critical for SecM stalling[14], whereas all other amino acids have primary amines with two hydrogens that would allow simultaneous hydrogen bonding with Ala164 as well as extraction of a proton by the 2′ OH of A76 (Fig. 7b). In our structure, Arg163 establishes a complex network of interactions with the ribosome, which we propose stabilizes the C-terminal end of the SecM NC and, in particular, the carbonyl-oxygen of Ala164 to interact with Pro166, thereby explaining why Arg163 is also critical for SecM-mediated stalling[14,22]. Lastly, our MD simulations indicate that the pulling force on the N-terminus of the SecM NC relieves stalling by disrupting the interaction between Ala164 and Pro166 (Fig. 6c), but that for this to occur, the α-helix of SecM must be unfolded first (Fig. 6b). Collectively, this suggests that the secondary structures, such as the α-helix observed in SecM, could act to fine-tune the efficiency of stalling by modulating the force required for relief of stalling.

The model presented here differs fundamentally from that based on a previous structure of SecM-SRC where the sidechain of Arg163

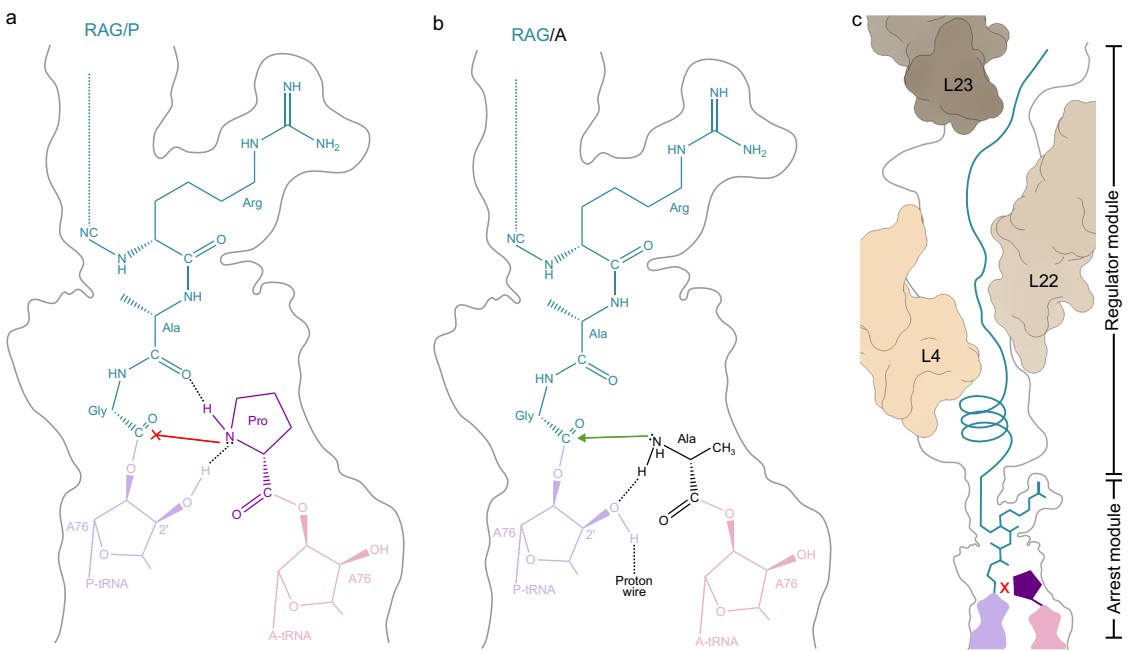

**Fig. 7 | Model for SecM-mediated translational arrest.** Schematic representation of the (**a**) RAG/P arrest module from SecM and (**b**) non-stalling RAG/A motif in the PTC. **c** SecM stalling is strongly driven by the RAG-P arrest module, however, the N-terminal regulator module also modulates and fine-tunes the stalling efficiency.

was proposed to extend into A-site and sterically block accommodation of the Pro-tRNA[24]. By contrast, in our SecM-SRC dataset, we observe no functional states with vacant A-sites, which is consistent with previous biochemical data showing that the SecM-stalled ribosome is unreactive to puromycin[12,13,17] and resilient to tmRNA rescue[13]. One possible explanation for this difference is the addition of chloramphenicol during purification of the previous SecM-SRC[24], which could have caused the loss of the A-site tRNA. In this regard, we note that SecM stalls with an alanine in the penultimate position of the SecM-NC attached to the P-site tRNA, which favours binding of chloramphenicol at the A-site[27,72]. Indeed, the binding site of chloramphenicol in the A-site overlaps with the position of the sidechain of Arg163 (Supplementary Fig. 10a–c). Other differences, such as the extended, rather than compacted, conformation of the SecM NC within the exit tunnel may have arisen due to the lower resolutions (3.7 Å and 6–9 Å) of the previous SecM-SRC structures[23,24] (Supplementary Fig. 10d–h).

While the structure of SecM-SRC and derived mechanism of stalling determined here differ with that of other ligand-independent arrest peptides, such as VemP[58] or MifM[73], we note a striking similarity with recently determined structures of phylogenetically-unrelated arrest peptides ApdA and ApdP[74]. Unlike SecM, which stalls at the C-terminal RAG/P motif[14], the ApdA and ApdP arrest peptides stall at a conserved RAP/P motif[75]. Nevertheless, the superimposition of the ApdA and ApdP with SecM illustrates a remarkable similarity in the conformation of the respective motifs (Supplementary Fig. 11a–f), supporting a common mechanism of peptide bond inhibition[74]. To provide additional support for the commonality in mechanism, we could also demonstrate that the RAG/P motif in SecM could be mutated to RAP/P and retained an equivalent level of stalling both in vivo and in vitro (Supplementary Fig. 11g, h). Collectively, our findings suggest that ApdA, ApdP and a range of other recently identified arrest peptides with RAG/P and RAP/P motifs from a range of diverse bacteria[75,76] are likely to utilize the same mechanism to induce translational stalling as described here for SecM.

Although the RAG/P motif of SecM plays a critical role in translational stalling, we observed defined interactions between other

residues of the SecM NC and components of the ribosomal tunnel and note that mutations in these regions can also influence the efficiency of translational arrest[14,18–20]. This leads us to expand our model for SecM-mediated arrest to comprise two modules, the RAG/P or "arrest module" that is attached to the P-tRNA and directly involved in preventing peptide bond formation together with a Pro-tRNA in the A-site, and a second N-terminal "regulator" module that can modulate the strength of stalling (Fig. 7c). We envisage that the regulator module could strengthen stalling by adopting secondary structures and/or establishing additional interactions with the ribosomal tunnel that increase the pulling force requirement to relieve stalling. However, we also envisage that in some cases, the regulator module may weaken or even prevent stalling by perturbing the fine-placement of the RAG/P motif (Fig. 6c), which would explain why RAGP motifs can also be found in non-stalling proteins[76]. Additionally, we believe that the N-terminal regulator module may be responsible for the species-specificity observed for SecM, where stalling is efficient on *E. coli*, but not *B. subtilis*, ribosomes[77]. The basis for this proposal is that the region around the PTC is highly conserved between *E. coli* and *B. subtilis* ribosomes, whereas the largest differences are observed within the tunnel, predominantly, within the ribosomal proteins uL4 and uL22.

## Methods

### Bacterial strains and plasmids
The protein coding sequence of SecM from *E. coli* was cloned into pDG1662 downstream of a T7 promoter, a ribosome binding site, a His-tag and a Flag-tag using restriction enzyme SphI and HindIII (NEB) and T4 ligase (NEB). The insert of SecM was amplified by PCR using Q5 High-Fidelity DNA polymerase (NEB) from *E. coli* strain K12 genomic DNA using primers Fwd_SphI_SecM (5′-TTTTTTGCATGCGTGAGTG-GAATACTGACG-3′) and Rev_SecM_stop_HindIII (5′-AAAAAAAAGC TTTTAGGTGAGGCGTTGAG-3′). DNA oligo primers used were purchased from Metabion.

### PCR and in vitro transcription
PCR reaction (Q5 High-Fidelity DNA Polymerase in Q5 Reaction buffer (NEB)) was used with primers M13 fwd (5′-GTAAAACGACGGCCAGT-3′)

and M13 rev (5′-CAGGAAACAGCTATGAC-3′) on the vector harbouring *secM* ORF to generate the amplified DNA sequence (5′-CAGGAAAC AGCTATGACCATGATTACGGAATTCGAGCTCGGTACCCGGGATCCCG CGAAATTAATACGACTCACTATAGGGGAATTGTGAGCGGATAACAAT TCCCCACTAGTAATAATTTTGTTTAACTTTAAGAAGGAGATATACC ATGGGCAGCAGCCATCATCATCATCATCACGATTACAAGGATGACGA CGATAAGGCTAGCAGCAGCGGTACCGGCAGCGGCGAAAACCTCTAT TTTCAGGGTAGTGCGCAAGCATGCGTGAGTGGAATACTGACGCGCT GGCGACAGTTTGGTAAACGTACTTCTGGCCGCATCTCTTATTAGG GATGGTTGCGGCGAGTTTAGGTTTGCCTGCGCTCAGCAACGCCGCCG AACCAAACGCGCCCGCAAAAGCGACAACCCGCAACCACGAGCCTTCA GCCAAAGTTAACTTTGGTCAATTGGCCTTGCTGGAAGCGAACACACG CCGCCCGAATTCGAACTATTCCGTTGATTACTGGCATCAACATGCCA TTCGCACGGTAATCCGTCATCTTTCTTTCGCAATGGCACCGCAAACA CTGCCCGTTGCTGAAGAATCTTTGCCTCTTCAGGCGCAACATCTTGC ATTACTGGATACGCTCAGCGCGCTGCTGACCCAGGAAGGCACGCCG TCTGAAAAGGGTTATCGCATTGATTATGCGCATTTTACCCCACAAGC AAAATTCAGCACGCCCGTCTGGATAAGCCAGGCGCAAGGCATCCGT GCTGGCCCTCAACGCCTCACCTAAAGCTTGGACTGGCCGTCGTTTT AC-3′; underlined are the T7 promoter region, ribosomal binding site, start codon, FLAG-tag and stop codon, respectively). PCR conditions applied were as suggested by the manufacturer and PCR products were purified via spin columns, and in vitro transcription reaction was set up using 1 μg PCR product per 50 μL reaction volume and T7 RNA polymerase (Thermo Scientific™). RNA was purified by LiCl precipitation and washed with ethanol.

### Generation of SecM-SRC

To generate the SecM-SRC, the transcribed template mRNA (250 ng μL$^{-1}$) was translated by incubation in an *E. coli* cell-free in vitro translation system (PURExpress® In Vitro Protein Synthesis Kit (NEB)). Briefly, a total reaction volume of 80 μL was prepared mixing 15.9 μL DEPC-treated water, 32 μL solution A, 24 μL solution B, 0.1 μL RNase Inhibitor (NEB) and 8 μL mRNA, and then incubated at 30 °C for 40 min with shaking in a thermomixer (500 rpm).

### Purification of the stalled-ribosomal complexes

The SecM-SRC was purified by incubating the in vitro translation reaction with 15 μL anti-FLAG® M2 affinity gel (Merck), previously equilibrated with Hico buffer (50 mM HEPES-KOH (pH 7.4, 4 °C), 100 mM potassium acetate, 15 mM magnesium acetate, 1 mM dithiothreitol, 0.01 % (w/v) n-dodecyl-beta-maltoside, sterile-filtered) inside a Mobicol column fitted with 35 μm filter (MoBiTec) at 4 °C for 3.5 h with rolling. After removal of the flow-through, the beads were washed with a total of 4 mL Hico buffer and then the bound complex was eventually eluted by incubation with 5 μL Hico buffer containing 0.6 mg mL$^{-1}$ 3XFLAG peptide for 45 min at 4 °C with rolling, followed by centrifugation (2000 × *g*, 4 °C, 2 min). Aliquots from each fraction were checked by western blotting or snap frozen and stored at −80 °C until needed.

### Cryo-EM sample preparation

3.5 μL of the SecM-SRC sample (8 OD$_{260}$/ml) were applied to grids (Quantifoil, Cu, 300 mesh, R3/3 with 3 nm carbon, Product: C3-C18nCu30-01) which had been freshly glow discharged using a Glo-Qube® Plus (Quorum Technologies) in negative charge at 25 mA for 30 s to make the grids hydrophilic. Sample vitrification was performed using mixture of ethane/propane in 1:2 ratio in a Vitrobot Mark IV (ThermoScientific), with the chamber set to 4 °C and 100% rel. humidity, and blotting performed for 3 sec with zero blot force with Whatman 597 blotting paper. The grids were subsequently clipped into autogrid cartridges and stored in liquid nitrogen until needed.

### Cryo-EM data collection

Data collection was performed on 300 kV Titan Krios G3i (Thermo Fisher/FEI) with Fringe-Free Imaging (FFI) setup and equipped with Gatan K3 direct electron detector using EPU (version 3.2.0.4775REL). Magnification of ×105,000 was used, with data collected using super resolution counted mode at 0.415 pixel size, binned twice on the fly through EPU yielding 0.83 pixel size. Total 40 e⁻/A² fluence was fractionated into 35 frames resulting in 1.14 e⁻/A² dose per frame and total exposure of 1.91 s in Nanoprobe mode (15 e⁻/px/s over an empty area on the camera level). Defocus range of −0.3 μm to −0.9 μm was used with step size of 0.1 μm between holes. C2 aperture of 70 μm was inserted with beam spot size of 7. BioQuantum energy filter set to 20 eV cut-off was used to remove inelastically scattered electrons. Final objective astigmatism correction <1 nm and auto coma free alignment <40 nm was achieved using AutoCTF function of Sherpa (version 2.11.1). A total of 4,388 micrographs were collected for SecM-SRC (12 exposures per hole) and saved as tiff gain corrected files.

### Single-particle reconstruction of SRC complexes

RELION v4.0[78,79] was used for processing, unless otherwise specified. For motion correction, RELION's implementation of MotionCor2 with 4 × 4 patches, and, for initial contrast transfer function (CTF) estimation, CTFFIND version 4.1.14[80], were employed. From 4,388 micrographs, 499,240 particles were picked using crYOLO with a general model[81]. In total, 398,692 ribosome-like particles were selected after two-dimensional (2D) classification and extracted at 2× decimated pixel size (1.66 Å per pixel) (Supplementary Fig. 1). An initial three-dimensional (3D) consensus refinement was done using a mol map based on *E. coli* 70S ribosome (PDB ID 7K00 with tRNAs and mRNAs removed), then initial 3D classification without angular sampling with five classes was performed. All 70S ribosomal like classes were combined (377,762 particles), followed by partial signal subtraction on the particles with a mask around tRNAs sites to perform focussed classification. One class containing 70S ribosomes with P-tRNA and A-tRNA (300,120 particles) was subsorted into four subclasses, of which one was of high resolution (300,107 particles); one class containing 70S with A/P hybrid state tRNA and P/E hybrid state tRNA (55,259 particles) was subsorted into four subclasses, of which one was of high resolution (36,489 particles). These two major classes were selected for further processing. In particular, the resulting classes´ subtracted particles were reverted to their original images and 3D refined and CTF refined (4$^{th}$ order aberrations, beam tilt, anisotropic magnification and per-particle defocus value estimation), then subjected to Bayesian polishing[82] and another round of CTF refinement. For the SecM-SRC with P-tRNA and A-tRNA a final resolution (gold-standard FSC$_{0.143}$) of masked reconstructions of 2.0 Å was achieved (Supplementary Fig. 1f, Supplementary Fig. 2a, b, e, f); for the SecM-SRC with A/P hybrid state tRNA and P/E hybrid state tRNA a final resolution (gold-standard FSC$_{0.143}$) of masked reconstructions of 2.6 Å was achieved (Supplementary Fig. 1g, Supplementary Fig. 3a). To estimate local resolution values, Bsoft[83] was used on the half-maps of the final reconstructions (blocres -sampling 0.83 -maxres -box 20 -cutoff 0.143 -verbose 1 -fill 150 -origin 0,0,0 -Mask half_map1 half_map 2) (Supplementary Fig. 2).

### Molecular modelling of the SRC complexes

The molecular models of the 30S and 5 S ribosomal subunits were based on the *E. coli* 70S ribosome (PDB ID 7K00)[84]. The tRNAs and nascent chains were modelled de novo. The secondary structure of the SecM nascent chain in the NPET was predicted using the PSIPRED 4.0 web service. Restraint files for modified residues were created using aceDRG[85], while the restraint file to link the tRNAs to their aminoacyl-/peptidyl- moiety was kindly provided by Keitaro Yamashita (MRC LMB, UK). Starting models were rigid body fitted using ChimeraX[86] and modelled using Coot 0.9.8.5[87] from the CCP4 software suite version 8.0[88]. The sequence for the tRNAs was adjusted based on the appropriate anticodons corresponding to the mRNA. Final refinements were done in REFMAC 5[89] using Servalcat[90]. The molecular models were

validated using Phenix comprehensive cryo-EM validation in Phenix 1.20–4487[91].

## β-galactosidase assay

*E. coli* cells (Supplementary Table 1) with plasmids (Supplementary Tables 2 and 3) were cultured in LB medium with 100 µg/mL ampicillin at 37 °C and withdrawn at an optical density at 600 nm ($OD_{600}$) of 0.5-1.0 for β-galactosidase assay. 100 µL portions of the cultures were transferred to individual wells of 96-well plate, and $OD_{600}$ was recorded. To lyse the cells, 50 µL of Y-PER reagent (Thermo Scientific) were added to the 100 µL of 10-fold diluted culture and the samples were frozen at −80 °C for at least 30 min. After thawing the samples, 30 µL of o-nitrophenyl-β-D-galactopyranoside (ONPG) in Z-buffer (60 mM $Na_2HPO_4$, 40 mM $NaH_2PO_4$, 10 mM KCl, 1 mM $MgSO_4$, 38 mM β-mercaptoethanol) was added to each well, $OD_{420}$ and $OD_{550}$ were measured every 5 min over 60 min at 28 °C. Arbitrary units [AU] of β-galactosidase activity were calculated by the formula [(1000 × $V_{420}$ − 1.3 × $V_{550}$)/$OD_{600}$], where $V_{420}$ and $V_{550}$ are the first-order rate constants, $OD_{420}$/min and $OD_{550}$/min, respectively.

## Bacterial in vitro translation arrest assay

In vitro translation arrest assay was carried out using *E. coli*-based coupled transcription-translation system (PUREfrex 1.0; GeneFrontier). 2.5 U/L of T7 RNA polymerase (Takara) was added further to reassure transcription. The DNA templates were prepared by PCR using primers and template DNA listed in Supplementary Table 4. After the translation reaction at 37 °C for 20 min, the reaction was stopped by adding three volumes of 1.3 × SDS-PAGE loading buffer (167 mM Tris-HCl (pH 6.8), 2.7% (wt/vol) SDS, 20% (vol/vol) glycerol, 6.7 mM DTT, a trace amount of bromophenol blue), and, when indicated, samples were further treated with 0.2 mg/ml RNase A (Promega) at 37 °C for 10 min to degrade the tRNA moiety of peptidyl-tRNA immediately before electrophoresis.

## Eukaryotic in vitro translation arrest assay

The DNA templates were prepared by PCR using primers and templates listed in Supplementary Table 4. In vitro transcription was carried out using T7 RNA Polymerase ver.2.0 (TaKaRa) and 150-250 ng of PCR product per 10 µl reaction volume. The mRNA was then purified by RNAClean XP (Beckman Coulter) and used for in vitro translation using the Rabbit Reticulocyte Lysate (RRL) translation system (Promega). A total reaction volume of 4 µL was prepared by mixing 2.8 µL Rabbit Reticulocyte Lysate (Nuclease-Treated), 10 µM Amino Acid Mixture Minus Methionine, and 10 µM Amino Acid Mixture Minus Leucine with the 75 nM mRNA. After the translation reaction at 30 °C for 20 min, the reaction was stopped by adding 24 volumes of SDS-PAGE loading buffer (125 mM Tris-HCl (pH 6.8), 2% (wt/vol) SDS, 15% (vol/vol) glycerol, 5 mM DTT, a trace amount of bromophenol blue), and, when indicated, samples were further treated with 0.1 mg/ml RNase A (Promega) at 37 °C for 20 min to degrade the tRNA moiety of peptidyl-tRNA immediately before electrophoresis.

## Western blotting

Samples were separated by 10% polyacrylamide gel prepared with WIDE RANGE Gel buffer (Nacalai Tasque), transferred onto a PVDF membrane, and then subjected to immuno-detection using antibodies against GFP (Wako, mFX75) or FLAG-tag (F3165; Sigma). Images were obtained and analyzed using Amersham Imager 600 (GE Healthcare) luminoimager. The band intensities were quantified using ImageQuant TL (GE Healthcare).

## Setup of MD simulations

The starting structure for the MD simulations was obtained by extracting, from the model of the SecM-SRC with A- and P-site tRNAs, all residues, water molecules, $K^+$, and $Mg^{2+}$ ions within 35 Å of the SecM NC. Pro166 was modelled as uncharged and with the α-amino hydrogen pointing towards the carboxylic oxygen of Ala164 as in reference[74]. The protonation states of the histidine residues were determined using the WHATIF software[92]. The structure was then placed in a triclinic orthogonal box, aligning the principal axes of the SecM peptide along the x,y,z coordinate axes. The longest axis of SecM was aligned with the z-axis and the minimum distance between the atoms and the box boundaries was set to 1.5 nm. To accommodate the pulling of the SecM residues out of the exit tunnel, the z-dimension of the box was extended by 2 nm in the pulling direction, resulting in simulation box dimensions of 18.60 nm, 13.25 nm, 12.83 nm. The system was then solvated with OPC water[93] using the programme solvate[94]. GENION[94] was used to add 7 mM $MgCl_2$ and 150 mM KCl and to neutralize with $K^+$ ions[94]. The ions were modelled using the $K^+$ and $Cl^-$ parameters from Joung and Cheatham[95] and the microMg parameters from Grotz et al.[96]. Partial charges of Pro166 were determined. The simulation system contained 402,745 atoms and 90,783 water molecules. All simulations were performed using GROMACS 2022[94] with the amber14sb forcefield[97]. Lennard–Jones and short-range electrostatic interactions were computed within a cut-off of 1 nm. Long-range electrostatic interactions were computed for distances larger than 1 nm using the particle-mesh Ewald summation[98] with a 0.12 nm grid spacing. Bond lengths were constrained using the LINCS algorithm[99] and virtual sites[100] were used for hydrogen atoms, allowing for an integration time step of 4 fs. The temperature coupling was performed using velocity rescaling[101] and solute and solvent were coupled independently to a heat bath at 300 K with a coupling time constant of 0.1 ps. Coordinates were recorded every 5 ps.

Firstly, the system was energy minimized with harmonic position restraints ($k = 1000$ kJ mol$^{-1}$ nm$^{-1}$) applied to the solute heavy atoms. After that, 8 replicas of the system were simulated for 70 ns to allow for solvent equilibration. During the first 50 ns, position restraints ($k = 1000$ kJ mol$^{-1}$ nm$^{-1}$) were applied on all the heavy atoms of the solute. During the following 20 ns, the position restraints were linearly decreased to zero for all the heavy atoms of the solute placed within 25 Å from the NC. Simultaneously, the force constant of the restraints applied to the heavy atoms positioned further than 25 Å from the NC was decreased to the one obtained from the fluctuations previously observed in full-ribosome simulations as described earlier[102]. Production runs (70-270 ns) were then carried out for 5 replica keeping the position restraints only on the outer-shell heavy atoms. During both equilibration steps and production run, the pressure was coupled to a stochastic cell rescaling barostat[103] with a time constant of 5 ps and scaling the box every 10 steps.

To investigate how stalling is relieved by pulling on the peptide, we carried out pulling MD simulations. To that aim, we added a harmonic potential, representing a spring, which depends on the distance $d$ and has a spring constant of 5000 kJ mol$^{-1}$ nm$^{-1}$. Here, $d$ is the z-component of the difference vector between the centre of mass (COM) of the N-terminal Pro132 backbone atoms and the spring position. The initial spring position was set to the Pro132 COM position in the starting structure. In the pulling simulations, the spring position was moved with constant velocity in the z-direction (along the tunnel axis) by 5.6 nm during the length of the simulation τ. To probe the effect of the velocity, we used different pulling times τ = 32 ns, 64 ns, 128 ns, 256 ns, 512, and 1024 ns, resulting in velocities ranging from 0.175 m/s to ~0.005 m/s. For each τ, we carried out 8 simulations started from the 8 structures obtained from the solvent equilibration.

## Analysis of MD simulations

First, all unbiased and all pulling trajectories were aligned using 23 S rRNA P-atoms. To check if the SecM α-helix remains stable during the unbiased simulations, we extracted peptide coordinates every 10 ns

and used DSSP[104] to obtain the secondary structure. For each simulation and each residue, we calculated the probability of being in an α-helical secondary structure and subsequently the mean values and standard deviations over all simulations. To obtain the mobility of SecM residues, we calculated backbone root mean square fluctuations (rmsf) for each residue and simulation, and calculated mean values and standard deviation over all simulations (Fig. 6a). For the pulling simulations, we extracted 640 structures equally spaced between 0 ns and τ and calculated the z-component of the backbone COM of each SecM residue and subtracted the initial value for G165 (Fig. 6b). For each extracted structure, we obtained the secondary structure using DSSP. For each simulation, we then recorded the time when the α-helix began unfolding, which was defined as the time from which the number of α-helix residues remained below 6, whereas the end of helix unfolding was defined as the earliest time from which no residue was found to be in an α-helical secondary structure. To obtain the time when an A164 shift occurs, we recorded the time from which on the A164 COM z-component is larger than 95% of the z-components obtained from all unbiased simulations. For each τ, the mean and standard deviations of the N-terminus z-positions at the time of the three events were calculated over all pulling simulations (Fig. 6c and Supplementary Fig. 9a). The maximum force from the harmonic potential acting on the N-terminus before helix unfolding, during helix unfolding, and between helix unfolding and the A164 shift were recorded for each simulation to investigate the dependence on the pulling time (Supplementary Fig. 9b).

## Figures

UCSF ChimeraX 1.6.1 was used to isolate density and visualize density images and structural superpositions. Models were aligned using PyMol version 2.5.5 (Schrödinger). Figures were assembled with Adobe Illustrator (latest development release, regularly updated) and Inkscape v1.3.

## Reporting summary

Further information on research design is available in the Nature Portfolio Reporting Summary linked to this article.

## Data availability

Micrographs have been deposited as uncorrected frames in the Electron Microscopy Public Image Archive (EMPIAR) with the accession codes EMPIAR-11758. Cryo-EM maps have been deposited in the Electron Microscopy Data Bank (EMDB) with accession codes EMD-18534 (*SecM-SRC with A- and P-site tRNA*), EMD-18590 (*SecM-SRC with hybrid A/P- and P/E-site tRNAs*). A molecular model has been deposited in the Protein Data Bank with accession code 8QOA (*SecM-SRC with A- and P-site tRNA*). Publicly available data used included PDB ID 1VY4, 1Y9J, 3CC2, 3JBU, 4WFN, 5LZV, 5NCO, 5NWY and 5A8L, 5JTE, 6XHV, 7K00, 7O19, 7RQE, 8CVK, 8QCQ and 8QBT, as well as EMDB ID EMD-1829. Source data are provided with this paper.

## Code availability

Initial coordinates, input files and output coordinates of the MD simulations, including raw data for the MD figures are publicly available on Zenodo (10.5281/zenodo.10492465).

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

## Acknowledgements

We would like to thank Keitaro Yamashita for help with refinement, Machiko Murata and Naoko Muraki for their technical support. This work was supported by the Deutsche Forschungsgemeinschaft (DFG) (grant WI3285/11-1 to D.N.W), JSPS Grant-in-Aid for Scientific Research (Grant No. 16H04788, 26116008, 20H05926, and 21K06053 to S.C, 19K16044, and 21K15020 to K.F), and under Germany's Excellence Strategy grant no. EXC 2067/1-390729940 (L.V.B.). We acknowledge financial support from the Open Access Publication Fund of Universität Hamburg. Cryo-EM data collection was performed at the Multi-User CryoEM Facility at the Centre for Structural Systems Biology, Hamburg, supported by the Universität Hamburg and DFG grant numbers (INST 152/772-1|152/774-1|152/775-1|152/776-1|152/777-1 FUGG).

## Author contributions

F.G. generated the SecM-SRC sample and H.S. prepared and screened the cryo-EM grids and collected high resolution data. F.G. and M.M. processed the cryo-EM data, as well as generated and refined the molecular models. H.P. helped with data processing, model building and refinement. K.F. and S.C. performed the biochemical stalling assays. S.G. and L.V.B. performed the MD simulations. D.N.W. and F.G. wrote the manuscript with input from all authors. D.N.W conceived and supervised the project.

## Funding

## Competing interests
The authors declare no competing interests.
