## [Peer Review File · Nature Communications]

The SecM arrest peptide traps a pre-peptide bond formation state of the ribosomeREVIEWER COMMENTS

Reviewer #1 (Remarks to the Author):

Gersteuer et al study the mechanism by which the arrest peptide, SecM, stalls 70S E. coli ribosomes. Despite its widespread use to study translation, how translational arrest is achieved is poorly understood. CryoEM analysis revealed two populations of stalled ribosomes: (a) A highly populated, pre-translocation state, with A-tRNA attached to Proline166 and P-tRNA attached to Gly165 and the growing nascent chain; (b) ratcheted ribosomes with A- and P-tRNAs shifted to A/P- and P/E-tRNAs respectively. The authors also investigate how a pulling force may relieve SecM stalling by using molecular dynamics simulations.

The authors present improved maps and structures which show the SecM peptide with new details, and these results supersede previous studies by Bhushan et al, 2011 and Zhang et al, 2015. Here, the authors suggest that the mechanism of stalling is peptide bond formation inhibition by the orientation of Pro-tRNA presence in the A-site, and also, in part, by the formation of a helix. The presence of compaction in the SecM peptide had previously been suggested in prior FRET studies by Woolhead et al, but no helical structure has been previously observed in other cryoEM structures. It is not very obvious to me that a nine-residue helix is clearly observable in this structure, and this observation appears to also contradict the mutagenesis studies presented in Woolhead et al. How the SecM stall which traps the ribosome in a pre-peptide bond state can then be overcome and permit ratcheting was also not clear.

Overall, this cryoEM study addresses an important set of questions in the field. However in its current form, it isn't obvious what advances have been made in the understanding the mechanism of SecM translational arrest relative to prior studies by e.g., Bhushan S et al which reported similar phenomena around the PTC. The maps and structures, although greatly improved relative to prior studies, may also need more evaluation (see below).

Comments to the authors:

1. The resolution for the main structure appears to be overestimated because the gradient of the FSC curve does not drop sharply as one would expect. Even the second structure which is lower in resolution, shows a more definitive FSC curve in Supp 2 and 3. The authors should revisit these results.
2. In the analysis of the structures where the helix is described and interactions are reported, please add the exact threshold you use. Also, how many standard deviations and/or the sigma level? The local resolution of the hybrid state should be reported too.

3. In Supp fig 3 and in the maps provided, the CCA motif looks good resolution but the nascent chain is less so, and there may be more than one population. Can you show codon-anticodon for your exact sequence? Can the authors rule out that there isn't additional complexity within the maps that needs to be resolved (e.g., multiple stalling events)?

4. Lines 195-196: can the authors clarify whether they have the resolution to make this statement regarding contacts with uL4?

5. Lines 201-205: Some additional clarity on how a 9-residue helix is observed is needed here, particularly with respect to the presence of a known helix breaker (Proline) and prior mutagenesis work (Woolhead et al). Woolhead et al have shown that mutation in the residues corresponding to this helix, including W155A, does not interfere with the compaction that they observed in the wild-type sequence (W155). Those authors suggest that side chain interactions are likely required for arrest. Can the authors further clarify their structural observations and relate these to Woolhead?

6. In relation to Point 6, the MD simulations also suggest that a complete 9-residue helix isn't populated to a significant extent. Could the simulations and Figure 6b include additional residues, such as Pro153 and the two N and C terminus of the helix for reference?

7. Lines 244-254: The extent of interactions that the authors observe should be revisited. For example, the side-chain of Tyr137 is not visible in fig 2 and 3 in the orientation presented, and generally the density around D140-132 is visible at lower threshold and in the main chain only.

8. Line 284: D140 is not placed in the density even at the lower threshold shown

9. Lines 288-303: if the local resolution is better than 8angstrom, please show it because the helix should be visible at this resolution

10. Water mediated interactions are not all convincing. W6+W8 have continuous density and not suggesting water rather than another additive in your buffers or noise. W14 density very unambiguous for water molecule

11. Sup 3 – minor typographical error in the figure

12. How do the maps and models compare to Bhushan et al?

Reviewer #2 (Remarks to the Author):

The manuscript by Gersteu*, Morici* et al. presents the structural basis of bacterial ribosome stalling by the SecM arrest peptide. This work is highly complementary to the companion manuscript submitted to the same journal in which there appears to be a unified mechanism of ribosome stalling by R-A-P/G-P peptides. The N-terminal “regulator module” is especially fascinating as it indicates that interactions within the exit channel can also modulate the extent of ribosome stalling. Much of the paper contends that previous mechanisms proposed by Zhang et al. (eLife, 2015) are likely artefactual. The more “native-like” conditions used in isolating the ribosomes for this paper (e.g., without chloramphenicol and using full-length SecM), as well as higher resolution, provide strong rationale in the authenticity of the present work.

The structural work is excellent. This paper strengthens and is strengthened by the other ApdA/ApdP paper. I recommend publication.

Reviewer #3 (Remarks to the Author):

This is a very interesting study that addresses the long-studied and controversial topic of SecM-induced stalling. The manuscript clearly articulates a range of conflicting data that has been presented throughout the years, and they report a new cryo-EM structure that indicates the nascent peptide forms a stable alpha helix within the ribosomal tunnel. This suggests that the presence of the helix distorts the positioning of the A-site tRNA, such that peptide bond cannot proceed. This is in contrast to other studies that have suggested A-site accommodation is impeded by the SecM peptide. To gain a more complete understanding of the mechanistic implications of the helical structure, the team applied all-atom explicit-solvent simulations. In their simulations, they applied a force to the tail of the peptide, in order to ask whether such a force can dislodge the trapped A-site tRNA tail. They also compared these pulling simulations to unperturbed simulations, which show the helix is stable in the tunnel. Through their simulations, they show that tRNA is likely to reorganize only after the formed alpha helix encounters a steric constriction, which requires the helix to unfold. Upon unfolding of the helix, the peptide may further extend, which allows force to be transferred to the 3'-CCA tail of the tRNA, thereby allowing it to organize and proceed with peptide bond formation.

With regard to the simulation aspects of the study, the text is very clear and the methods are suitable. The team used well-established techniques appropriately, and their simulations provide compelling evidence for the mechanistic influence of helix formation in the tunnel.

Reviewer #4 (Remarks to the Author):

The manuscript by Gersteuer F. et al. "The SecM arrest peptide adopts a helical conformation and traps a pre-peptide bond formation state of the ribosome" reports a high-resolution (2.0 Å) structure of a translating E. coli 70S ribosome stalled by the SecM leader peptide. SecM, the secretion monitor arrest peptide, is located upstream of the secA gene and regulates the expression of the SecA ATPase that resides in the cell membrane. SecA functions with the SecYEG translocon to facilitate insertion into and through the cell membrane. The SecM peptide has an N-terminal signal region that is recognized by SecA residing in the cytoplasmic membrane. However, translation of SecM is problematic for the ribosome, inducing stalling by presumably interfering with peptide-bond formation. In the absence or low level of SecA, the ribosome stalls while translating the SecM leader peptide, which modulates the conformation of the downstream mRNA structure exposing the Shine-Dalgarno region of the secA gene, allowing its translation. In contrast, when SecA is present in the cell membrane, the signal region of the SecM peptide interacts with SecA and the pulling force exerted by SecA rescues the stalled ribosome and the ribosome-binding site of the secA gene remains unavailable.

The structural basis by which SecM stalls the ribosome remains unclear. A previous cryo-EM study of the E. coli 70S ribosome stalled with SecM lacked the A-site tRNA, and it was proposed that the SecM peptide interferes with the accommodation of the incoming Pro-tRNA(Pro). These results were at odds with biochemical data reporting that the SecM stalled ribosome complex contains the Pro-tRNA(Pro) in the ribosomal A site. The structure could not rationalize why Pro166 in the A site of the ribosome is required for the SecM-mediated stalling, while a mutation to alanine relieves stalling. An additional discrepancy from the previous structure is the presence of an extended SecM nascent peptide chain in the ribosome peptide exit tunnel (NPET), contrary to the proposed compact conformation based on mutational and FRET studies. Thus, the molecular basis of SecM-mediated ribosome stalling has remained obscure.

This study illuminates this by providing a detailed structure of the E. coli 70S stalled with the SecM peptide. Here, the authors use the full-length SecM (170 residues), which allows to visualize additional interactions between SecM and elements of the ribosome exit tunnel beyond the previously described SecM stalling motif 150FxxxWlxxxGIRAG165/P166. Furthermore, the presented structure was obtained in the absence of a translation inhibitor, such as chloramphenicol used previously (Zhang et al. 2015). The authors traced and modeled 34 amino acids (Pro132 to Gly165) of the SecM peptide in the

NPET of the ribosome. Furthermore, clear density is seen for the A-site Pro-tRNA(Pro), enabling building of the most accurate model of the SecM-stalled ribosome to date.

The high-resolution of the EM map allows the description of many interactions between the SecM peptide and the wall of the NPET, including rRNA nucleotides and the loop of uL22. The structure also allows the authors to propose a compelling model for how the SecM-stalled peptide interferes with peptide bond formation. First, it is obvious from this structure that the SecM peptide did not transfer onto the Pro-tRNA(Pro), despite the fact that the nucleotides of the peptidyl transferase center (PTC) are in the known induced conformation. Correspondingly, clear density is observed for the proline residue attached to the A-site tRNA. The authors propose that the carbonyl oxygen of Ala164, which acts as an acceptor of hydrogen bonds, would sequester the only hydrogen on Pro166, and the 2'OH of A76 of P-site tRNA would then act as a donor of H-bond that makes possible the H-bonding interaction between Pro166 and Ala164 of SecM. These interactions decrease the nucleophilicity of Pro166, which would usually be created by the 2'OH of A76 through proton extraction from the A-site amino acid. Also, the interaction with the carbonyl of Ala164 creates a geometry that is not conducive for a nucleophilic attack on the carbonyl-carbon of Gly165 on the P-site tRNA. Because proline is the only residue that has a secondary amine with only one hydrogen, this model rationalizes the requirement for proline in the A site of the stalled SecM ribosome complex.

The authors further performed molecular dynamic simulations to mimic a pulling force applied to the SecM peptide by SecA, allowing to elucidate how it can relieve translational stalling. The pulling force, applied to Pro132, gradually unfolds the α -helix of SecM in the upper region of the NPET and shifts Ala164 breaking its interaction with Pro166, which then allows Pro166 to do the nucleophilic attack on Gly165 to transfer the SecM peptide to Pro-tRNA(Pro) in the A site.

Overall, the data is of excellent quality and the high-resolution of the EM map allows the visualization of SecM-mediated ribosome stalling with unprecedented details. It clarifies the mechanism of ribosome stalling by SecM, which has been used as a model ribosome stalling peptide in numerous studies. In my view, this manuscript could be published as is. Nevertheless, I have the following minor comments and suggestions that may improve the quality and accuracy of the presentation.

In essence, the current structure is not the first that visualizes a SecM-stalled ribosome with a bound A-site Pro-tRNA. Thus, only referring to the 2015 structure (Zhang et al., 2015) that is lacking the A-site substrate and not mentioning previous studies is somewhat misleading. For instance, in the introduction and discussion, the authors do not adequately present their previous model (Bhushan et al, 2011). In this study, two populations of ribosomes were described, ratcheted and unratcheted. A subpopulation of the unratcheted ribosomes had an additional density for the A-site tRNA. The authors attributed the partial occupancy of the A-site Pro-tRNA_{Pro} to its partial dissociation from the A site caused by high-salt washes used during the purification. Granted, however, the resolution of this reconstruction was low, at 9.3 Å, which impeded a description of the details presented here. The authors could mention this study in the

introduction and bring it back in the discussion section explaining why they were able to capture the current complex – i.e. possibly because of the different purification procedure used.

Line 128: “stalled ribosomal complexes...” The authors present one structure here at a resolution of 2.0 Å.

Line 422: “...for peptide bond formation to be formed.” Repeated wording.

In the legend of Suppl. Fig. 7: “(e) Native confirmation...” conformation.

Same “(h) Overlay...in non-bonded confirmation...”

Same in title of Suppl. Fig. 8:

Supplementary Figure 8 SecM does not stall eukaryotic ribosome in vitro and confirmation of ribosomal PTC components.

In same figure legend, “(b)...The cording region...” coding

In title of Suppl. Fig. 11:

Supplementary Figure 11 Comparison of RAG/P motive... motif

Matthieu Gagnon

REVIEWER COMMENTS

Reviewer #1 (Remarks to the Author):

Gersteuer et al study the mechanism by which the arrest peptide, SecM, stalls 70S E. coli ribosomes. Despite its widespread use to study translation, how translational arrest is achieved is poorly understood. CryoEM analysis revealed two populations of stalled ribosomes: (a) A highly populated, pre-translocation state, with A-tRNA attached to Proline166 and P-tRNA attached to Gly165 and the growing nascent chain; (b) ratcheted ribosomes with A- and P-tRNAs shifted to A/P- and P/E-tRNAs respectively. The authors also investigate how a pulling force may relieve SecM stalling by using molecular dynamics simulations.

The authors present improved maps and structures which show the SecM peptide with new details, and these results supersede previous studies by Bhushan et al, 2011 and Zhang et al, 2015. Here, the authors suggest that the mechanism of stalling is peptide bond formation inhibition by the orientation of Pro-tRNA presence in the A-site, and also, in part, by the formation of a helix. The presence of compaction in the SecM peptide had previously been suggested in prior FRET studies by Woolhead et al, but no helical structure has been previously observed in other cryoEM structures. It is not very obvious to me that a nine-residue helix is clearly observable in this structure, and this observation appears to also contradict the mutagenesis studies presented in Woolhead et al. How the SecM stall which traps the ribosome in a pre-peptide bond state can then be overcome and permit ratcheting was also not clear.

We are surprised that it is not obvious to the reviewer that a helix is clearly observable in the structure...we specifically included many figures showing the molecular model fitted into the density e.g. Figure 1e and 2b. We hope that the reviewer agrees that quality of the electron density is excellent and the model is unambiguous for this region – its actually one of the most well-resolved parts of the nascent chain. We also provided all maps and models and would ask the reviewer to look at the experimental data directly if not convinced by the manuscript figures. We also show the potential hydrogen bonds that are formed between the backbones of the $i+4$ and i residues, which define the geometry of an alpha-helix. Retrospectively, one could have also predicted the presence of an alpha-helix based simply on secondary structure predictions – they are surprisingly accurate. This is also shown in Figure 2c. This finding is in excellent agreement with the FRET studies of Woolhead et al. Since we do not have a structure of mutated versions of SecM, we cannot say whether the mutant versions of SecM form alpha-helices or not.

Overall, this cryoEM study addresses an important set of questions in the field. However in its current form, it isn't obvious what advances have been made in the understanding the mechanism of SecM translational arrest relative to prior studies by e.g., Bhushan S et al which reported similar phenomena around the PTC. The maps and structures, although greatly improved relative to prior studies, may also need more evaluation (see below).

We are surprised that it is not obvious to the reviewer what the advances are that have been made relative to the prior studies by e.g. Bhushan et al. The previous structures from Bhushan et al were at 6-9 Å and therefore the SecM nascent chain could not be modelled with any confidence. Indeed, like Zhang et al, Bhushan et al proposed an extended conformation for the SecM nascent chain, whereas at higher resolution we observe that part of SecM adopts an

alpha-helical structure. We have now included additional images in Sup Fig 10 to illustrate this point. As mentioned in the manuscript, this completely changes the register of the SecM peptide and as a consequence a completely different set of interactions with components of the ribosomal tunnel is observed. Moreover, Bhushan et al conclude that the mechanism of inhibition is due to a shift in the P-site tRNA away from the Pro-tRNA in the A-site. As highlighted in Figure 5, we suggest a completely different mechanism where the carbonyl-oxygen of Ala164 of the SecM nascent chain in the P-site forms a hydrogen bond with hydrogen of the nitrogen of Pro166 on the A-site tRNA, which prevents the nucleophilic attack and ultimately short-circuits the peptidyltransferase activity of the ribosome.

Comments to the authors:

- 1. The resolution for the main structure appears to be overestimated because the gradient of the FSC curve does not drop sharply as one would expect. Even the second structure which is lower in resolution, shows a more definitive FSC curve in Supp 2 and 3. The authors should revisit these results.**

We have recently published ribosome structures with a range of resolutions, ranging from 1.6-2.2 Å (Paternoga et al NSMB 2022) and the density features seen here for SecM are fully consistent with the reported resolution of 2.0 Å. The map versus model FSC curves that were supplied in Sup Fig 2 do not support the conclusion that there is any overfitting.

- 2. In the analysis of the structures where the helix is described and interactions are reported, please add the exact threshold you use. Also, how many standard deviations and/or the sigma level? The local resolution of the hybrid state should be reported too.**

The density for the helix is shown in Fig 2b so we have now added the threshold used for this image to the legend for panel 2b, as requested. We have not included the local resolution for the hybrid state in Sup Fig. 3 because the density is fragmented and the local resolution for is thereby biased by the high resolution of the surrounding rRNA and gives misleading results i.e. it suggests higher resolution than it actually is.

- 3. In Supp fig 3 and in the maps provided, the CCA motif looks good resolution but the nascent chain is less so, and there may be more than one population. Can you show codon-anticodon for your exact sequence? Can the authors rule out that there isn't additional complexity within the maps that needs to be resolved (e.g., multiple stalling events)?**

The reviewer is correct, the density for the tRNA and NC are not well-resolved in the hybrid state, therefore, for this reason we cannot use the codon-anticodon density to ascertain the sequence and identity of the tRNAs. We specifically wrote on page 7, lines 182-183: "we cannot exclude that this population represents a mixture of states, which coupled with the poor resolution of NC, meant that state was not analyzed further." However, further subsorting of this population did not yield any additional states, but it is still possible that the similarity of the tRNAs prevents similar states from being distinguished from one-another.

- 4. Lines 195-196: can the authors clarify whether they have the resolution to make this statement regarding contacts with uL4?**

We observe no density, even at low thresholds, that connects the SecM NC with uL4, suggesting that if there are interactions, they are no stable ones. We cannot rule out transient

interactions. Hence, we wrote “The SecM NC makes no stable contact with uL4 as it passes through the constriction”. We hope the reviewer agrees that this statement is appropriate.

5. Lines 201-205: Some additional clarity on how a 9-residue helix is observed is needed here, particularly with respect to the presence of a known helix breaker (Proline) and prior mutagenesis work (Woolhead et al).

While it is true that Proline residues are generally considered helix breakers, there is evidence that when located at the N-terminus of the helix that they can actually facilitate helix formation. We now mention this on page 7 and have added a new reference to support this statement. We also note here that we have changed the text to read a 7-residue helix because of the reviewer’s comments in point 6 below. The basis for assigning a 9-residue helix was that Ala159 and Ser151 at either end of the helix have backbones that can hydrogen bond with the residue four places ahead of it, however, we have now noticed that ChimeraX displays only helix ribbons for residues T152-Q158 (as seen in Figure 2f), thus yielding a 7-residue helix, which is consistent with that displayed in the MD simulations (Fig. 6a,b). Even with the 7-residue helix, the Pro153 is still part of the helix as defined by DSSP and ChimeraX (which uses DSSP parameters to define the helices to be displayed).

Woolhead et al have shown that mutation in the residues corresponding to this helix, including W155A, does not interfere with the compaction that they observed in the wild-type sequence (W155). Those authors suggest that side chain interactions are likely required for arrest. Can the authors further clarify their structural observations and relate these to Woolhead?

Our results are consistent with Woolhead et al to the extent that a W155A mutation would be compatible with the formation of the alpha-helix observed in our structure. Without prior knowledge, we would not predict that a W155A mutation would relieve SecM-mediated translational arrest, therefore, we do not think our structure can explain this finding from Woolhead et al. Instead, a structure of the W155A variant-stalled ribosome would be required, which is not trivial since the variant does not stall. One also has to remember that a change in the sequence (such as W155A) can cause a change in the path of the nascent polypeptide chain as it is being synthesized so that even if the same helix forms in the tunnel with W155A variant, the helix may be located completely differently to interfere with the placement of the RAPP motif to prevent stalling. Although interesting, the explanations are very speculative and due to limited space, we have not included them in the manuscript.

6. In relation to Point 6, the MD simulations also suggest that a complete 9-residue helix isn’t populated to a significant extent.

Actually, the MD simulations show that the helix observed in the cryo-EM structure is very stable and remains populated through-out the simulations. The discrepancy arises because the assigned of the nine-residue helix in the cryo-EM structure was initially based on the ability of the backbone residues to hydrogen bond with residues located four places ahead or behind them and thus included Ser151 and Ala159, generating a 9-residue helix. By contrast, the assignment of the helices in the MD simulations is determined by the DSSP algorithm that assigns the alpha-helix to residues Thr152 to Glu158, i.e. a 7-residue helix. We note here that ChimeraX (which also uses the DSSP algorithm) also displays a 7-residue and not 9-residue helix, as seen in Figure 2f. Therefore, we have now adjusted the text in the manuscript to refer to the helix as 7 rather than 9 residues and we thank the reviewer for bringing our attention to this discrepancy.

Could the simulations and Figure 6b include additional residues, such as Pro153 and the two N and C terminus of the helix for reference?

For reference, we have now drawn the C α atoms of the N- and C-terminal residues of the α -helix (T152 and Q158) as spheres in Fig. 6b.

7. Lines 244-254: The extent of interactions that the authors observe should be revisited. For example, the side-chain of Tyr137 is not visible in fig 2 and 3 in the orientation presented, and generally the density around D140-132 is visible at lower threshold and in the main chain only.

We have revisited the interactions as requested by the reviewer. The reviewer is quite correct that the quality of the N-terminal region is not as good as the central and C-terminal regions. Nevertheless, we do observe clear density for Tyr137 but not at the threshold shown in Fig 2 and 3 because we used a higher threshold to ensure that the density for the rRNA and tRNA does not dominate the image. However, for this reason, we have included an image of the nascent chain at lower threshold in Sup Fig 2, where one can see the density for this residue. Related to point 8 below, we also agree that the density for D140 is not convincing and have decided to remove this interaction from the figure and text.

8. Line 284: D140 is not placed in the density even at the lower threshold shown

We agree that the density for the sidechain of D140 is not great. Aspartates often suffer from radiation damage. To be more conservative, we have removed this interaction from the figure 3a and removed Asp140 interaction from the associated text on page 10.

9. Lines 288-303: if the local resolution is better than 8angstrom, please show it because the helix should be visible at this resolution

The helix is only observed at very low thresholds, suggesting quite some flexibility. Therefore, we have not displayed local resolution since it is biased by the high resolution of the surrounding rRNA and gives misleading results i.e. it suggests higher resolution than it actually is.

10. Water mediated interactions are not all convincing. W6+W8 have continuous density and not suggesting water rather than another additive in your buffers or noise. W14 density very unambiguous for water molecule

We have re-examined the water molecules and agree that W6 and W8 may not be waters because in filtered maps they merge somewhat, perhaps suggesting that they may be a polyamine – not that we have polyamines in the buffer, but we cannot rule out the possibility that they purified with the ribosome. For this reason, we have removed the interactions of W6 and W8 from the text as well as Figure 4d. By contrast, we have retained W14 (now termed W12) since we believe the density and interactions are consistent with a water molecule.

11. Sup 3 – minor typographical error in the figure

We have corrected the typographical error of “Non-rotaed” to “Non-rotated”. Thank you for pointing this out.

12. How do the maps and models compare to Bhushan et al?

The maps of Bhushan et al are 6-9 Å and therefore do not have sidechain information, hence, we excluded them from the analysis. In fact, Bhushan did not even deposit a model in the PDB therefore we can only compare our model with their density. This is now included in the revised Sup Fig. 10 where one can see that the cryo-EM density of SecM nascent chain has very little features – certainly no sidechain information and one can debate about whether the path is similar or different to that reported here.

Reviewer #2 (Remarks to the Author):

The manuscript by Gersteur*, Morici* et al. presents the structural basis of bacterial ribosome stalling by the SecM arrest peptide. This work is highly complementary to the companion manuscript submitted to the same journal in which there appears to be a unified mechanism of ribosome stalling by R-A-P/G-P peptides. The N-terminal “regulator module” is especially fascinating as it indicates that interactions within the exit channel can also modulate the extent of ribosome stalling. Much of the paper contends that previous mechanisms proposed by Zhang et al. (eLife, 2015) are likely artefactual. The more “native-like” conditions used in isolating the ribosomes for this paper (e.g., without chloramphenicol and using full-length SecM), as well as higher resolution, provide strong rationale in the authenticity of the present work.

The structural work is excellent. This paper strengthens and is strengthened by the other ApdA/ApdP paper. I recommend publication.

We thank the reviewer for their very positive assessment and agree that it is very complementary to the ApdA/ApdP paper.

Reviewer #3 (Remarks to the Author):

This is a very interesting study that addresses the long-studied and controversial topic of SecM-induced stalling. The manuscript clearly articulates a range of conflicting data that has been presented throughout the years, and they report a new cryo-EM structure that indicates the nascent peptide forms a stable alpha helix within the ribosomal tunnel. This suggests that the presence of the helix distorts the positioning of the A-site tRNA, such that peptide bond cannot proceed. This is in contrast to other studies that have suggested A-site accommodation is impeded by the SecM peptide. To gain a more complete understanding of the mechanistic implications of the helical structure, the team applied all-atom explicit-solvent simulations. In their simulations, they applied a force to the tail of the peptide, in order to ask whether such a force can dislodge the trapped A-site tRNA tail. They also compared these pulling simulations to unperturbed simulations, which show the helix is stable in the tunnel. Through their simulations, they show that tRNA is likely to reorganize only after the formed alpha helix encounters a steric constriction, which requires the helix to unfold. Upon unfolding of the helix, the peptide may further extend, which allows force to be transferred to the 3'-CCA tail of the tRNA, thereby allowing it to organize and proceed with peptide bond formation.

With regard to the simulation aspects of the study, the text is very clear and the methods are suitable. The team used well-established techniques appropriately, and their simulations provide compelling evidence for the mechanistic influence of helix formation

in the tunnel.

We thank the reviewer for the positive assessment of the molecular dynamics simulations part of the study. We also agree that they nicely complement the structure to provide compelling evidence for the mechanism of SecM stalling, as well as the influence of the helix in relief of the arrest.

Reviewer #4 (Remarks to the Author):

The manuscript by Gersteuer F. et al. “The SecM arrest peptide adopts a helical conformation and traps a pre-peptide bond formation state of the ribosome” reports a high-resolution (2.0 Å) structure of a translating E. coli 70S ribosome stalled by the SecM leader peptide. SecM, the secretion monitor arrest peptide, is located upstream of the secA gene and regulates the expression of the SecA ATPase that resides in the cell membrane. SecA functions with the SecYEG translocon to facilitate insertion into and through the cell membrane. The SecM peptide has an N-terminal signal region that is recognized by SecA residing in the cytoplasmic membrane. However, translation of SecM is problematic for the ribosome, inducing stalling by presumably interfering with peptide-bond formation. In the absence or low level of SecA, the ribosome stalls while translating the SecM leader peptide, which modulates the conformation of the downstream mRNA structure exposing the Shine-Dalgarno region of the secA gene, allowing its translation. In contrast, when SecA is present in the cell membrane, the signal region of the SecM peptide interacts with SecA and the pulling force exerted by SecA rescues the stalled ribosome and the ribosome-binding site of the secA gene remains unavailable.

The structural basis by which SecM stalls the ribosome remains unclear. A previous cryo-EM study of the E. coli 70S ribosome stalled with SecM lacked the A-site tRNA, and it was proposed that the SecM peptide interferes with the accommodation of the incoming Pro-tRNA(Pro). These results were at odds with biochemical data reporting that the SecM stalled ribosome complex contains the Pro-tRNA(Pro) in the ribosomal A site. The structure could not rationalize why Pro166 in the A site of the ribosome is required for the SecM-mediated stalling, while a mutation to alanine relieves stalling. An additional discrepancy from the previous structure is the presence of an extended SecM nascent peptide chain in the ribosome peptide exit tunnel (NPET), contrary to the proposed compact conformation based on mutational and FRET studies. Thus, the molecular basis of SecM-mediated ribosome stalling has remained obscure.

This study illuminates this by providing a detailed structure of the E. coli 70S stalled with the SecM peptide. Here, the authors use the full-length SecM (170 residues), which allows to visualize additional interactions between SecM and elements of the ribosome exit tunnel beyond the previously described SecM stalling motif 150FxxxxWxxxxGIRAG165/P166. Furthermore, the presented structure was obtained in the absence of a translation inhibitor, such as chloramphenicol used previously (Zhang et al. 2015). The authors traced and modeled 34 amino acids (Pro132 to Gly165) of the SecM peptide in the NPET of the ribosome. Furthermore, clear density is seen for the A-site Pro-tRNA(Pro), enabling building of the most accurate model of the SecM-stalled ribosome to date.

The high-resolution of the EM map allows the description of many interactions between the SecM peptide and the wall of the NPET, including rRNA nucleotides and the loop of

uL22. The structure also allows the authors to propose a compelling model for how the SecM-stalled peptide interferes with peptide bond formation. First, it is obvious from this structure that the SecM peptide did not transfer onto the Pro-tRNA(Pro), despite the fact that the nucleotides of the peptidyl transferase center (PTC) are in the known induced conformation. Correspondingly, clear density is observed for the proline residue attached to the A-site tRNA. The authors propose that the carbonyl oxygen of Ala164, which acts as an acceptor of hydrogen bonds, would sequester the only hydrogen on Pro166, and the 2'OH of A76 of P-site tRNA would then act as a donor of H-bond that makes possible the H-bonding interaction between Pro166 and Ala164 of SecM. These interactions decrease the nucleophilicity of Pro166, which would usually be created by the 2'OH of A76 through proton extraction from the A-site amino acid. Also, the interaction with the carbonyl of Ala164 creates a geometry that is not conducive for a nucleophilic attack on the carbonyl-carbon of Gly165 on the P-site tRNA. Because proline is the only residue that has a secondary amine with only one hydrogen, this model rationalizes the requirement for proline in the A site of the stalled SecM ribosome complex.

The authors further performed molecular dynamic simulations to mimic a pulling force applied to the SecM peptide by SecA, allowing to elucidate how it can relieve translational stalling. The pulling force, applied to Pro132, gradually unfolds the α -helix of SecM in the upper region of the NPET and shifts Ala164 breaking its interaction with Pro166, which then allows Pro166 to do the nucleophilic attack on Gly165 to transfer the SecM peptide to Pro-tRNA(Pro) in the A site.

Overall, the data is of excellent quality and the high-resolution of the EM map allows the visualization of SecM-mediated ribosome stalling with unprecedented details. It clarifies the mechanism of ribosome stalling by SecM, which has been used as a model ribosome stalling peptide in numerous studies. In my view, this manuscript could be published as is. Nevertheless, I have the following minor comments and suggestions that may improve the quality and accuracy of the presentation.

We thank the reviewer for their time and efforts and their positive evaluation of the manuscript.

In essence, the current structure is not the first that visualizes a SecM-stalled ribosome with a bound A-site Pro-tRNA. Thus, only referring to the 2015 structure (Zhang et al., 2015) that is lacking the A-site substrate and not mentioning previous studies is somewhat misleading. For instance, in the introduction and discussion, the authors do not adequately present their previous model (Bhushan et al, 2011). In this study, two populations of ribosomes were described, ratcheted and unratcheted. A subpopulation of the unratcheted ribosomes had an additional density for the A-site tRNA. The authors attributed the partial occupancy of the A-site Pro-tRNA_{Pro} to its partial dissociation from the A site caused by high-salt washes used during the purification. Granted, however, the resolution of this reconstruction was low, at 9.3 Å, which impeded a description of the details presented here. The authors could mention this study in the introduction and bring it back in the discussion section explaining why they were able to capture the current complex – i.e. possibly because of the different purification procedure used.

We agree that we could have introduced the Bhushan et al paper more extensively. This was not done because the low resolution of the study meant there were too many limitations in the

interpretation. However, we agree that it would be appropriate to put an additional sentence before introducing the Zhang et al study to maintain the historical record, which we have now done on page 4, lines 90-93. We would also point out that Bhushan et al did not deposit a molecular model in the PDB so we cannot compare molecular models. Nevertheless, we have generated an image for the cryo-EM density for the SecM nascent chain from Bhushan et al which is now included in Sup Fig. 10 showing that it is rather featureless and not sufficiently resolved to assign sidechains. We leave it to the reviewer's imagination as to whether there is any similarity in the path of the nascent chain compared to that reported here.

Line 128: “stalled ribosomal complexes...” The authors present one structure here at a resolution of 2.0 Å.

Corrected

Line 422: “...for peptide bond formation to be formed.” Repeated wording.

Replaced “formed” with “attained”

In the legend of Suppl. Fig. 7: “(e) Native confirmation...” conformation.
Same “(h) Overlay...in non-bonded confirmation...”

Corrected

Same in title of Suppl. Fig. 8:

Supplementary Figure 8 SecM does not stall eukaryotic ribosome in vitro and confirmation of ribosomal PTC components.

In same figure legend, “(b)...The cording region...” coding

Corrected

In title of Suppl. Fig. 11:

Supplementary Figure 11 Comparison of RAG/P motive... motif

Corrected

Matthieu Gagnon

REVIEWERS' COMMENTS

Reviewer #1 (Remarks to the Author):

The authors have considered all comments and concerns, and it was a pleasure to read the revised version of the paper. This work will be a high-quality addition to the ribosome field, helping us to understand how and why ribosomes arrest at high resolution.

Reviewer #2 (Remarks to the Author):

The revised manuscript by Gersteuer*, Morici* et al. presents a more thorough analysis of SecM-mediated stalling in *E. coli* ribosomes. There are many advances from this paper that extend or alter our understanding from previous work, most notably the actual stalling mechanism (unfavorable chemistry and geometry by the A-site proline). Other aspects of this paper enhance our understanding of the stalling mechanism, including the compacted nascent chain conformation and its interactions with ribosome exit tunnel components. The structure work is compelling and of high quality. I have no further comments about this manuscript and strongly recommend publication.